# Human motion recognition and prediction using loose cloth

Tianchen Shen [1] ✉, Sacha Morris [1], Irene Di Giulio [2] & Matthew Howard [1] ✉

Human motion analysis plays a crucial role in fields such as healthcare, human-robot interaction and virtual reality. Conventional approaches typically rely on tightly attached body sensors, which can prove uncomfortable and impractical. Here, we investigate motion recognition and prediction using garments incorporating embedded sensors. We analyse how the movement of loose-fitting clothing can predict body motion in both simulated and real-world scenarios. Results demonstrate that sensors attached to fabric can improve recognition accuracy by up to 40% improvement and require approximately 80% less movement history compared to sensors directly attached to the body. These findings indicate that garment motion provides valuable information for analysing human movement. The study additionally offers insights regarding the design of intelligent textiles with integrated sensing capabilities.

The wearable sensors market is expanding rapidly, with a compound annual growth rate of 18.3% anticipated, rising from USD 840 million in 2021 to USD 3.7 billion by 2030[1]. This rapid growth is driven by their numerous applications in areas such as (i) medical diagnosis, monitoring and therapy[2] (e.g. targeted management of conditions such as Parkinson's disease and stroke[3–5]), (ii) fitness and wellbeing (e.g. monitoring biomarkers such as steps taken, heart rate, and muscle oxygen saturation[6]), (iii) entertainment[7] (e.g. providing immersive and interactive experiences in video games and e-sports[8]), and (iv) manufacturing (e.g. monitoring behaviour in the work place to reduce the incidence of accidents, enhance productivity[9] or provide new human-machine interfaces[10]).

These applications are built on the ability to capture, recognise, and predict human behaviours in real time[11–14]. Current motion capture technologies, such as marker-based optical motion capture[15], wearable sensor-based systems (e.g. accelerometers/inertial measurement units (IMUs)[16]), and computer vision-based motion capture[17], have significant limitations[18]. For instance, systems are often not user-friendly[19], as they require sensors or markers to be taped to the body, causing discomfort or irritation upon removal. Additionally, they involve complex setups, where sensors or markers may need to be precisely positioned, a process that requires expert anatomical knowledge and is particularly time-consuming when a large number of sensors are involved[20]. Furthermore, computer vision-based motion capture systems are often constrained by their environmental

requirements, needing specific lighting conditions or specific surroundings (e.g. retro reflective surfaces for infra-red cameras) that limit potential applications[21,22].

Inspired by the latest e-textiles technology, which now allows sensors to be embedded in clothing[23–25], this work proposes the use of sensor-embedded garments to capture motion and analyse human behaviour. The proposed motion capture system in this work ensures comfortable, easy-to-use, and unobtrusive wear for users[24] and allows for the recording of human movement outside the laboratory for long periods[26]. A comparison between different types of motion capture systems is presented in Supplementary Note 1. It highlights the advantages of sensor-embedded garments in terms of factors such as comfort and ease of use. However, a challenge in recording human movement using e-textiles is the potential inclusion of motion artefacts caused by the movement of the clothing with respect to the body[27]. Many different approaches have been used to address this issue. For example, (i) sensors tightly attached to rigid components of the body (e.g. arm, thigh, wrist, etc.)[28–30], (ii) attached to tight-fitting clothing[16,31–33] or (iii) statistical machine learning/signal processing methods have been employed to reduce artefacts[34–38]. In contrast, a limited body of work[39–41] suggests that the information contained in fabric motion can be used for human activity recognition (AR). Therefore, this paper focuses on predicting human motion patterns from loose clothing. To the authors' knowledge, this problem has not been previously investigated.

[1]Department of Engineering, King's College London, London, England, UK. [2]School of Basic & Medical Biosciences, King's College London, London, England, UK. ✉e-mail: tianchen.shen@kcl.ac.uk; matthew.j.howard@kcl.ac.uk

In this paper, the term motion recognition is used to refer to the process of identifying which human motion is occurring (e.g. walking or running) based on past movement, while motion prediction refers to predicting the future time course of a movement (e.g. the positions of a point of the body over time). Figure 1 illustrates the problem and the approach taken in this study: future movement is predicted by first rapidly recognising its class from the movement of the clothing from the recent past set of sensor readings, then using a model to predict future human motion. The hypothesis is that motion recognition accuracy achieved with the analysis of clothing movement surpasses that attained through body movement and requires a significantly shorter duration of past movement data to make a correct recognition. The basis of this hypothesis is that clothing features a flexible structure that supports multi-directional movement, enhancing its range and complexity[42]. This, in turn, amplifies the distinction between two classes of movement, consequently, simplifying the detection of the movement class. In this study, left-to-right hidden Markov models (LR-HMMs) are applied to form models from motion capture data based on measurement of orientation, although the findings are replicated with different statistical modelling methods and with different modalities of sensing (not reported here). The motivations for this chosen study design are that (i) LR-HMMs are well-suited for modelling time series data across various applications[43] due to their ease of construction, manipulation, and the existence of optimal algorithms for training and scoring, such as the forward algorithm and the Viterbi algorithm[44], and (ii) measurement of orientation is the most common modality of sensing found in untethered motion capture systems as it is the primary sensing modality provided by accelerometers/IMUs[45]. The proposed approach works by eliciting the movement class from the LR-HMMs, then using those same LR-HMMs to predict the movement. To understand the reason why using fabric movement can get higher motion recognition accuracy than using rigid movement, the statistical distances (i.e. cross-fitness distance) between two LR-HMMs for both rigid and fabric movement are computed to estimate discrimination information between two classes of movement.

In summary, this research offers the following main contributions. (i) It explores how complex motions can be predicted using sensors mounted on loose clothing. (ii) It compares the performance of motion recognition and prediction when using tightly- and loosely-attached sensors for prediction tasks of varying difficulty. (iii) It reports experiments in predicting human reaching, a common daily activity. The results provide insights into the counterintuitive phenomena involved in using sensorised loose clothing, and how it can offer

additional valuable information for the analysis of human motion. This research challenges the conventional wisdom that motion prediction requires sensors to be attached tightly to the body. Combined with the advantages of e-textiles, such as unobtrusive sensing and enhanced user comfort, this approach not only broadens the scope of motion capture systems but also ensures greater user acceptance. It paves the way for advancements in the design and deployment of motion capture technologies, using ordinary clothing to achieve improved accuracy and ease of use. Moreover, the broader implications of these findings may lead to new insights in biology, anthropology, and the social sciences. Animals and humans commonly have or make use of loose appendages and garments. For instance, spiders sense their prey via faint vibrations in their web[46,47], while humans employ flowing attire in cultural displays to augment their expressiveness[42]. The finding that there is often increased, useful information content in the motion of loose appendages suggests nature may already be exploiting this effect for evolutionary and socio-cultural advantage.

## Results

### Movements of different frequency predicted more accurately and efficiently with fabrics

One of the simplest possible movement analysis tasks is to recognise and predict one-dimensional motion patterns across movements of varying frequency. When determining which movement is occurring, the larger the difference in frequency, the lower the difficulty of the prediction task. As a simple test-bed for this, an instrumented scotch yoke with a piece of fabric attached is utilised to physically implement simple harmonic motion[48] with different frequencies by driving the yoke at different speeds. The position and yaw orientation of the yoke are detailed in Fig. 2, while the sensor readings are shown in Fig. 3a. The movements of the scotch yoke and fabric can be found in Supplementary Video 1.

The motion recognition accuracy using the past movement of the rigid-attached sensor $R_1$ and the fabric-attached sensor $F_4$ from the first time $t = 0$ s to $t = 0.25$ s for each difficulty of prediction task is stated in this Fig. 3b. From this, the rigid movement is predicted (shown as the dashed line and shaded area). The dashed line is the most likely future trajectory. The shaded area represents the probability of the future trajectory (specifically, the further away from the dashed line, the less likely the trajectory is to occur).

The motion recognition accuracy from each sensor evaluated based on their movements from the initial time $t = 0$ s to various times up to $t = 2.5$ s are shown in Fig. 3c. As observed, the sensor $F_4$

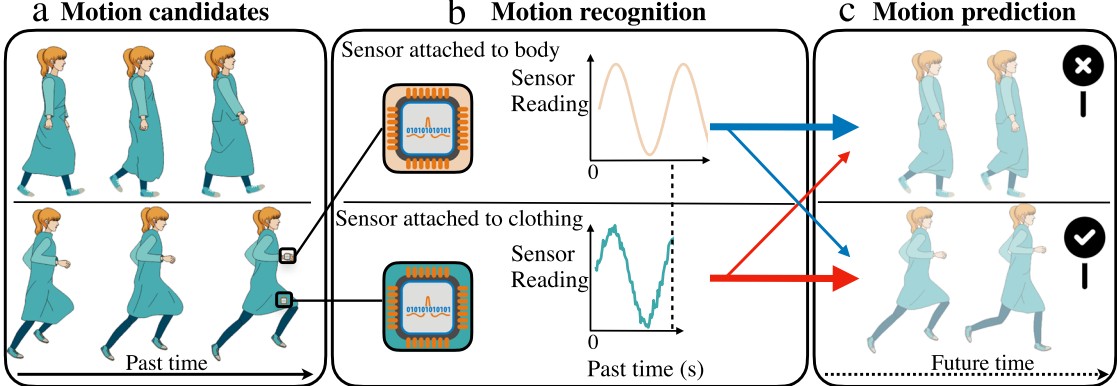

**Fig. 1 | Stages of human movement recognition and prediction. a** Motion candidates: One sensor attached to the body and another one attached to clothing simultaneously capture movement data, which are then used to independently identify a specific human motion between candidates (i.e. walking and running). **b** Motion recognition: Once calibrated, the model can be used to recognise human motion based on readings from the fabric sensor alone.

This avoids the user having to don uncomfortable and inconvenient body-attached sensors. It also achieves higher motion recognition accuracy than would be seen in a system using body-attached sensors alone. **c** Motion prediction: Future body movement (correct motion: running) is predicted based on feeding readings from the clothing-attached sensors into the statistical model.

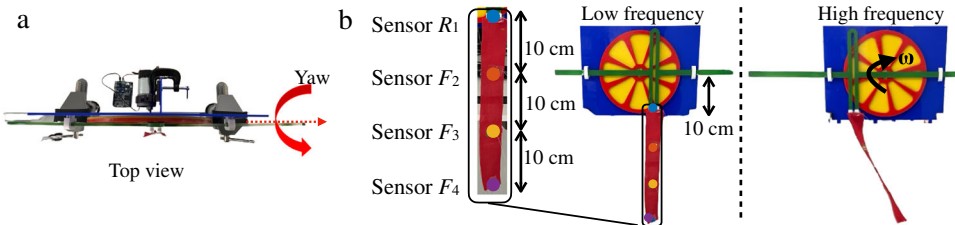

**Fig. 2 | Experimental setup for movements of different frequencies. a** The top view of the experimental setup. The angle (in red) shows the yaw orientation of rigid and fabric movement. **b** The front view of the experimental setup. The scotch yoke moves at different frequencies.

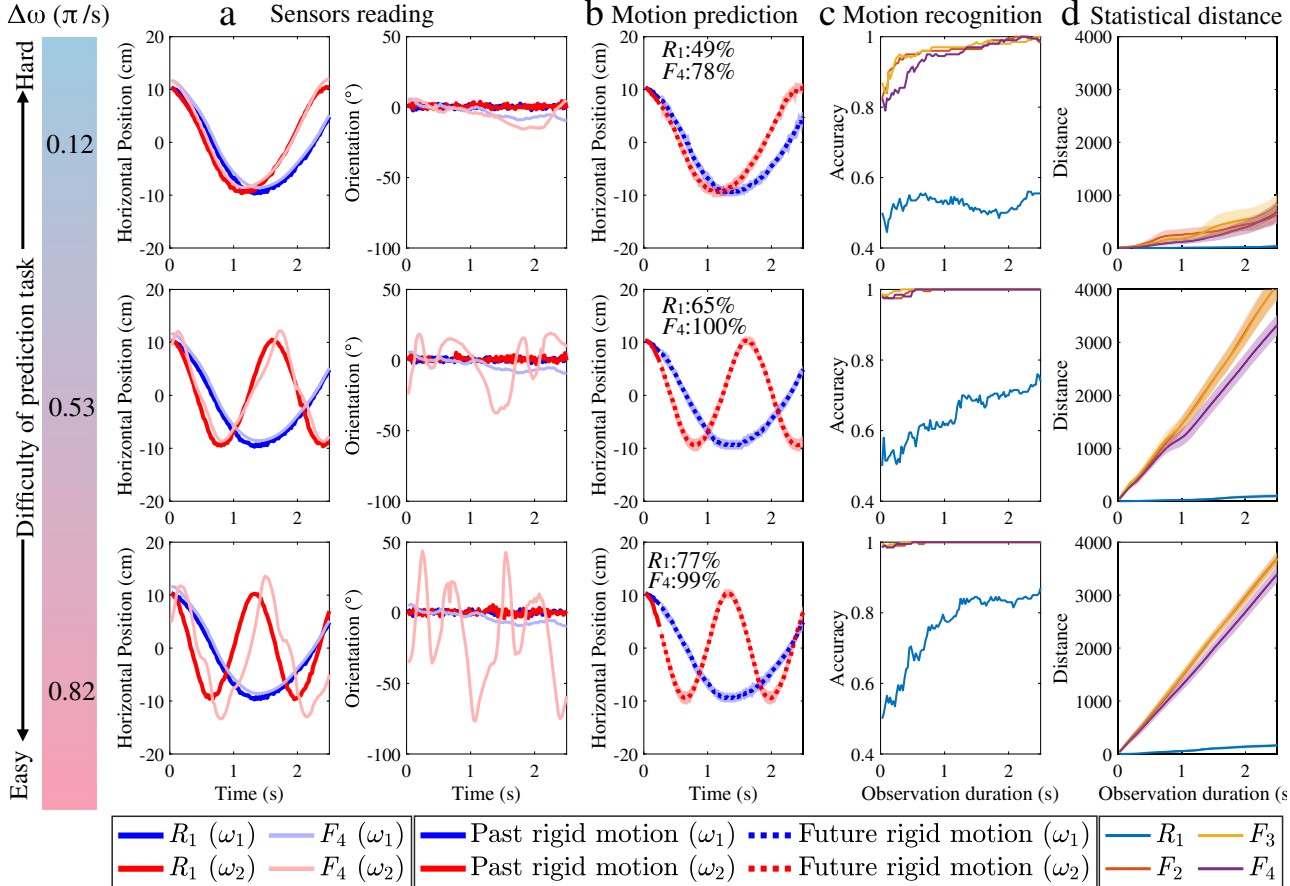

**Fig. 3 | Movement frequency predictions are more accurate and efficient using sensorised fabric. a** The actual horizontal moving trajectory and the yaw orientation of the sensor ($R_1$ thick and dark line) and the sensor ($F_4$ thin and light line). **b** The predicted rigid body motion (mean value dashed line; ± s. d. × 5 is shaded area) and the motion recognition accuracy of sensor $R_1$ and $F_4$ given the trajectory from the initial time to 0.25 s. **c** The motion recognition accuracy and **d** the statistical distances (mean value solid line; ± s. d. is shaded area) of each sensor give the past movement from the initial time to various times up to 2.5 s.

significantly outperforms the sensor $R_1$ in prediction tasks of varying difficulty. The accuracy of the sensor $F_4$ is significantly higher, and requires notably less time to achieve similarly high recognition accuracy compared to the sensor $R_1$ (i.e. $\Delta\omega = 0.12\pi$rad s$^{-1}$, higher rows in Fig. 3c). The advantage becomes less evident as the frequency difference between the two movements increases. For instance, the accuracy of the sensor $F_4$ only marginally higher and requires up to slightly less time compared to the sensor $R_1$ when the prediction task is easier (i.e. $\Delta\omega = 0.82\pi$rad s$^{-1}$, lower rows in Fig. 3c). The placement of fabric-attached sensors (i.e. $F_2$–$F_4$) does not appear to have an obvious effect on motion recognition accuracy.

To understand why fabric-attached sensors have higher motion recognition accuracy, the information content of the rigid and fabric movement is investigated. Figure 3d shows the cross-fitness distance

of each sensor between the LR-HMMs of two frequencies given the movement from the first time $t = 0$ s, across various time points up to $t = 2.5$ s. The cross-fitness distance of fabric-attached sensors is significantly higher than the rigid-attached sensor. This suggests that motion artefacts from fabric movement provide discriminative information, thereby facilitating the prediction task. The cross-fitness distance of clothing-attached sensors remains higher, irrespective of their positions in relation to the point of attachment of the fabric to the yoke. The corresponding analysis for prediction tasks of a variety of difficulties (i.e. $\Delta\omega = \{0.12\pi, 0.29\pi, 0.47\pi, 0.53\pi, 0.72\pi, 0.82\pi$rad s$^{-1}\}$) can be referred to Supplementary Note 2. To avoid potential intrinsic bias in the left-to-right hidden Markov model (LR-HMM) model, alternative statistical modelling techniques (e.g. temporal convolutional network and transformer-based neural networks) were

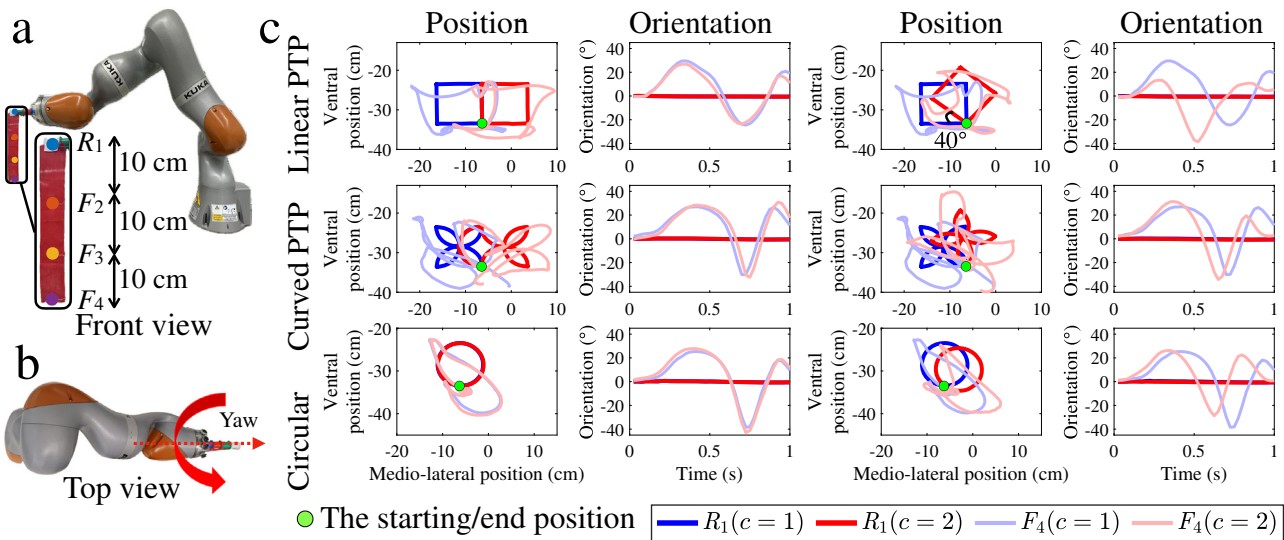

**Fig. 4 | Experimental setup for discrete movement patterns. a** The front and **b** top view of the experimental setup. The angle shows the yaw orientation of rigid and fabric movement. **c** The actual translational position and the yaw orientation of the sensors $R_1$ (thick and dark line) and $F_4$ (thin and light line) when the moving velocity is 22.5 cm s$^{-1}$. Note that the variations of rigid movements in all types of movements are less obvious.

employed to replicate the analysis. As detailed in Supplementary Note 3, these approaches produced comparable findings.

The above experiments were performed in a laboratory environment with ambient temperature and humidity, and without any wind. It is crucial to investigate the impact of real-world environmental conditions on motion prediction for loose-fitting clothing. A multi-functional climate control unit (3-in-1 air cooler, purifier and humidifier, Sealey UK) was employed to systematically control three key variables: airflow velocity, room temperature and relative humidity.

The motion recognition accuracy of fabric-mounted sensors exhibits slight degradation under low airflow velocities (i.e. 0.3 m/s), particularly during complex prediction tasks (i.e. $\Delta\omega = 0.47\pi rad\, s^{-1}$). However, at higher airflow velocities (i.e. 0.47 m/s), textile-based motion sensing performance demonstrates marked deterioration relative to rigid motion. In contrast, simple prediction tasks (i.e. $\Delta\omega = 0.82\pi rad\, s^{-1}$) maintain consistent accuracy across airflow conditions, suggesting minimal environmental susceptibility for fundamental motion analysis.

The motion recognition accuracy of fabric motion exhibits no statistically significant variation across a range of ambient temperatures (15–25 °C) and relative humidity levels (45–65%), as quantified through controlled environmental trials. Please refer to Supplementary Note 4 for more detailed results.

This experiment was also repeated using IMUs. The motion recognition performance for both rigid and fabric motion was evaluated using three-axis acceleration data from the sensors. The results can be found in Supplementary Note 5, which demonstrates that accelerometer data also support these findings.

### Discrete multidimensional movement patterns are predicted more accurately and efficiently with fabrics

Many movement analysis problems require the recognition and prediction of discrete (i.e. non-periodic), multi-dimensional movements of varying complexity. To evaluate the role of fabric movement in such settings, a robotic manipulator is used to generate movement in different patterns. This can be considered a proxy for human arm movement, but with the benefit that it produces highly precise and repeatable movements, thereby allowing the effect of fabric motion to be studied experimentally without the potential confounding effects of natural human movement variability (experiments involving actual human movement are reported in the subsequent section, below). In

this experiment, the task is to discriminate between movement patterns, the difficulty of which task depends on the difference in trajectory. To control the difficulty through a single variable, trajectories of a given shape are varied by rotating them around a single axis. The larger the angle of rotation, the easier the prediction task. The movements of the robot arm and fabric can be found in Supplementary Videos 2 and 3. The trajectory and the yaw orientation of sensors $R_1$ and $F_4$ (see Fig. 4a, b) in both the difficult ($\Delta\theta = 0°$) and easier ($\Delta\theta = 40°$) prediction tasks are shown in Fig. 4c. The sensor readings for more difficulties of prediction tasks can be found in Supplementary Note 6.

Figure 5a states the motion recognition accuracy using the past movement of sensor $R_1$ and $F_4$ from the first time $t = 0$ s to $t = 1.5$ s. From this prediction, the future end-effector movement (indicated by a dashed line and shaded area) is shown. The recognition accuracy for each sensor using different durations of past movement is also evaluated. Figure 5b illustrates the motion recognition accuracy, based on the past movement from the first time $t = 0$ s, across various time points up to $t = 1.5$ s. As can be seen, fabric-attached sensors have higher motion recognition accuracy than the rigid-attached sensor when the prediction task is challenging. Sensors with both attachments have very high accuracy when the prediction task is straightforward. The improved accuracy of the fabric-attached sensors remains largely consistent across different sensor placements.

Figure 5c shows the cross-fitness distances between two LR-HMMs given the trajectories between the first time $t = 0$ s, across various time points up to $t = 1.5$ s. The bigger cross-fitness distance indicates greater information content for discriminating between movements which simplifies the prediction task. This indicates that fabric movement provides additional information to simplify the prediction task. Supplementary Notes 7–9 show the results for prediction tasks with an increased variety of difficulties when the robot arm performs linear, curved point-to-point (PTP) and circular movements.

### Fabrics moving at greater speed exhibit greater information content

Repeating the study described in the preceding section with varying movement speeds reveals more about the relationship between speed and the information content of the fabric. Figure 6a illustrates the mean motion recognition accuracy between two classes of movement from time $t = 0$ s to different times until $t = 0.25$ s, for linear PTP movement at different rotation degrees when the robot's peak speed is

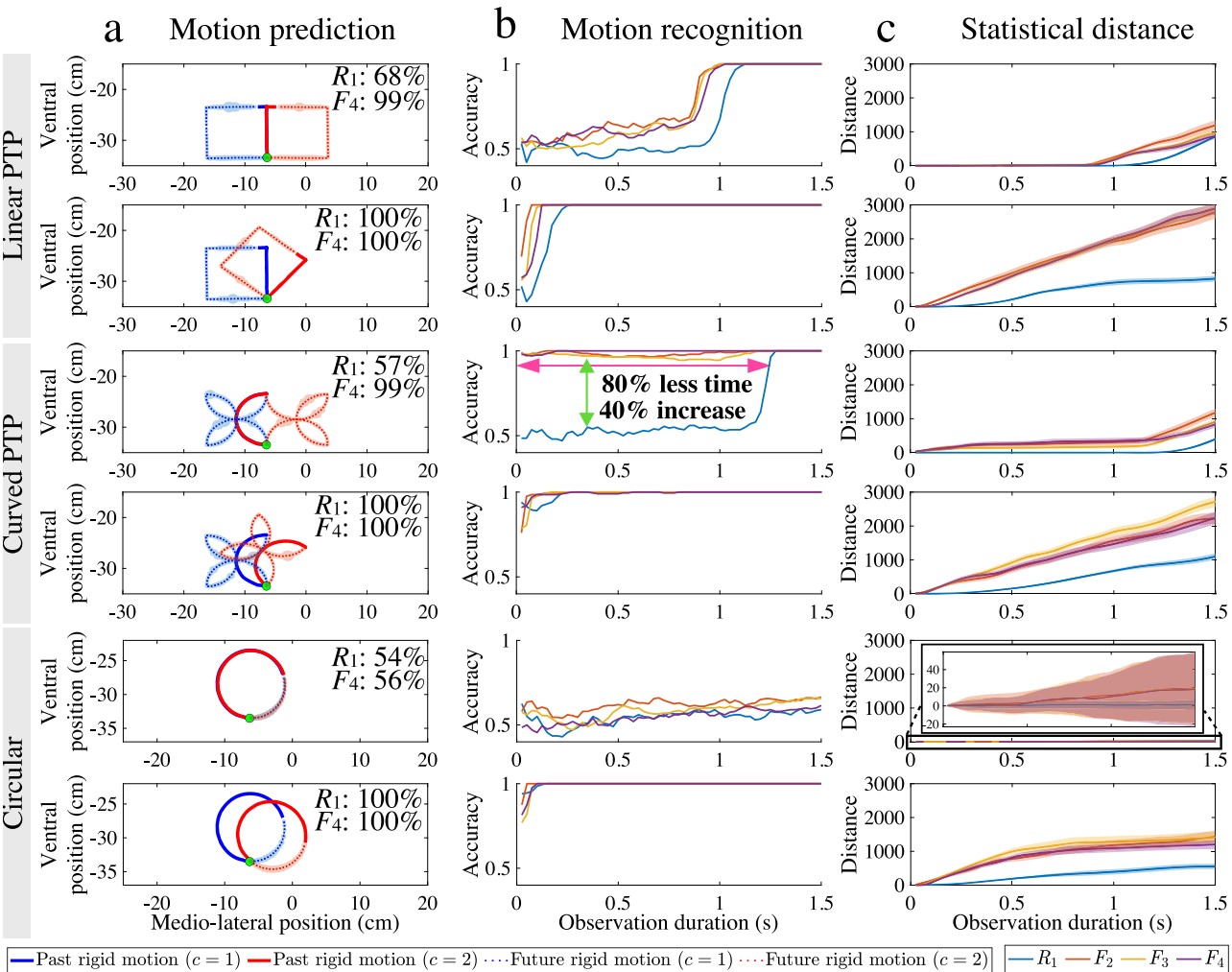

**Fig. 5 | Discrete movement patterns can be predicted more accurately and efficiently with sensorised fabric. a** The predicted rigid-body motion (mean value dashed line; ±s.d. × 5 is shaded area) and the motion recognition accuracy of sensor $R_1$ and $F_4$ given the trajectory from the initial time to 1 s. **b** The motion recognition accuracy and **c** the statistical distances (mean value solid line; ±s.d. is shaded area) of each sensor give the past movement from the initial time to various times to 1.5 s.

25 and 15 cm s⁻¹. Figure 6b–e shows the detailed motion prediction accuracy between two classes of movement from the first time $t = 0$ s to the time $t = 1.5$ s when the rotations $\Delta\theta$ are 0° and 90° and moves at different speeds. The results for curved PTP and circular movement can be referred to Supplementary Note 10. Fabric-attached sensors have significantly higher recognition accuracy than the rigid-attached sensor when the prediction task is challenging (e.g. the rotation $\Delta\theta$ is 0°) and the moving velocity (i.e. 25 cm s⁻¹) is higher. However, when the prediction task is less difficult (e.g. the rotation $\Delta\theta$ is 90°), there is no significant difference in the accuracy of both attachment types. The comparison of fabric movements at different moving speeds can be found in Supplementary Video 4. As can be observed, fabric has more wobble when the robot arm moves at 25 cm s⁻¹ compared to when the robot arm moves slower (i.e. 15 cm s⁻¹). This suggests that the behaviour of the fabric becomes more influential at higher movement speeds, resulting in fabric movement containing more additional information. This accounts for the observed higher motion recognition accuracy when the robot arm operates at a higher velocity. The results of different types of motion are shown in Supplementary Note 10.

**Human reaching is predicted more accurately from loose, sensorised garments**

The final study reported here tests whether the findings discussed so far translate to predicting human motion from loose garments. For

this, prediction in the context of reaching is studied as it is a common daily activity. Specifically, the scenario considered here consists of reaching to press one of a set of buttons, and the task is to predict which button the person intends to hit before the button is pressed. In this situation, the difficulty of the prediction task varies according to the distance between the target buttons since this determines the trajectory that the participant takes from the starting location (see Fig. 7). As the target buttons are equidistant from the starting location, this can be measured from the angular difference $\Delta\theta$. The participant's human reaching motion can be found in Supplementary Video 5. The vertical positions of both the wrist-attached sensor ($R$) and the sleeve-attached sensor ($F$) during the task of reaching target buttons with varying angular differences are shown in Fig. 8a. An Extended description of human reaching motion is shown in Supplementary Note 11.

Figure 8b illustrates the corresponding motion prediction results. Additionally, it states the motion recognition accuracy of sensors $R$ and $F$ given the past movement from $t = 0$ s to $t = 0.5$ s. As observed, the sleeve-attached sensor ($F$) shows higher motion recognition accuracy compared to the wrist-attached sensor ($R$). When the two target buttons are positioned further apart (easier prediction task), the improvement in accuracy is lower. Figure 8c, d shows the motion recognition accuracy and the cross-fitness distance for various task difficulties (i.e. $\Delta\theta = 5°$, 10°, 15°) based on the past movement from

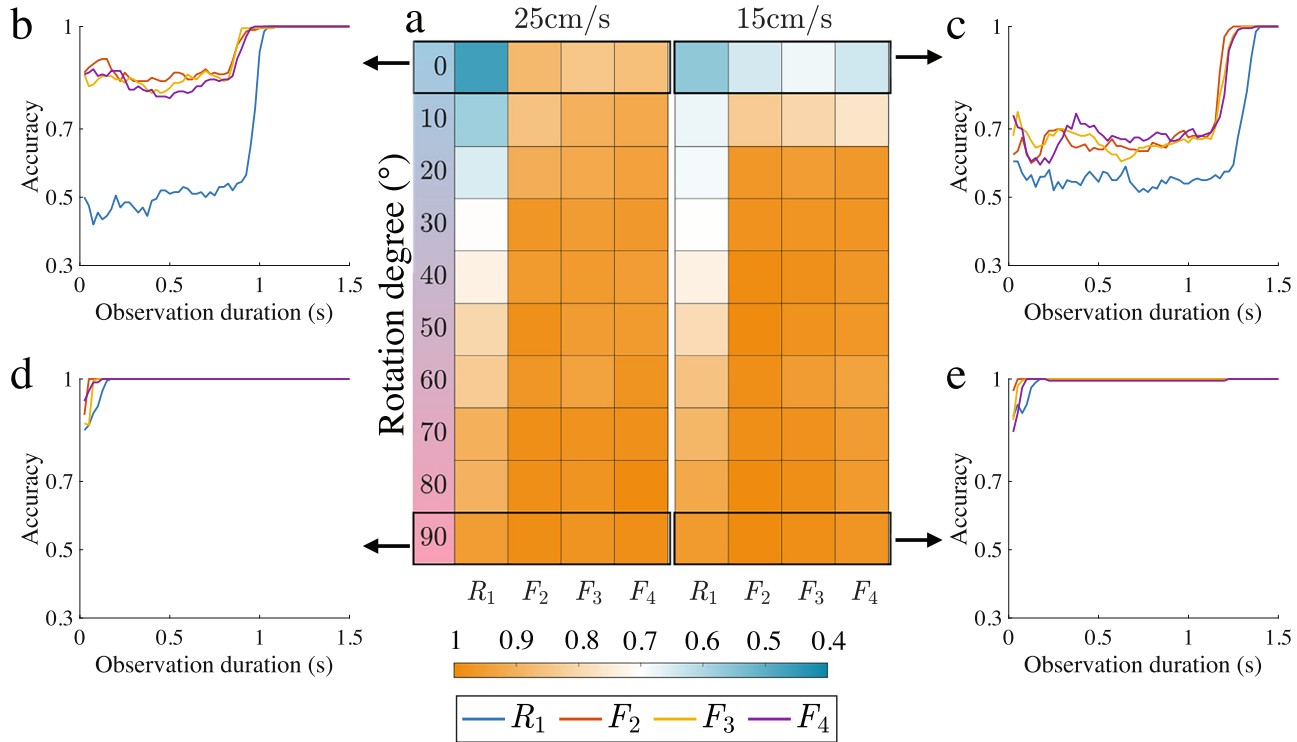

**Fig. 6 | Speed increases the information content of fabric motion. a** The heatmap shows the mean motion recognition accuracy given the trajectory from the initial time to 0.25 s across various rotation degrees in the contexts of linear PTP movement when the end effector of the robot arm is moving at 25 and 15 cm s⁻¹. The motion recognition accuracy of each sensor given the past movement from the initial time to various times to 1.5 s when the robot arm is moving at **b** 25 cm s⁻¹ and **c** 15 cm s⁻¹ for the harder prediction task (i.e. $\Delta\theta = 0°$). The motion recognition accuracy of each sensor given the past movement from the initial time to various times to 1.5 s when the robot arm is moving at **d** 25 cm s⁻¹ and **e** 15 cm s⁻¹ for the easier prediction task (i.e. $\Delta\theta = 90°$).

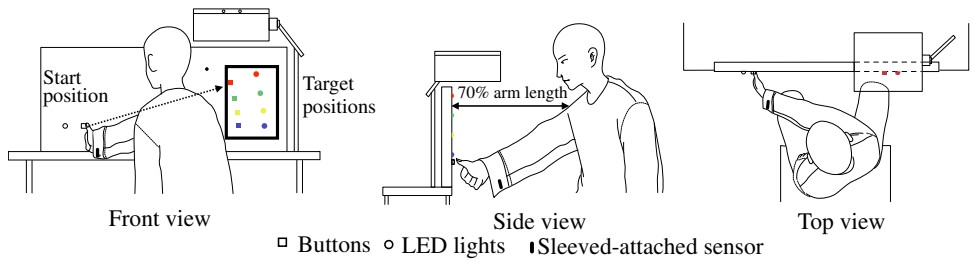

**Fig. 7 | Experimental setup for human reaching motion.** A schematic diagram of the experimental setup with different views.

$t = 0$ s, across various time points up to $t = 1.25$ s for all participants. In the case of the smallest angular difference (i.e. 5°), the accuracy of predictions from the sleeve-attached sensor surpasses that of the wrist-attached sensor, and the former requires less time to achieve high motion recognition accuracy. As the prediction task becomes easier (i.e. the angle increases), these advantages are reduced.

According to a Wilcoxon matched-pairs signed-rank test, the difference in the accuracy of motion recognition between using wrist-attached and sleeve-attached sensors during the initial phase (i.e. from $t = 0$ s to $t = 0.125$ s, $p > 0.01$) is not statistically significant. This may be attributed to the relatively low arm moving speed at the beginning of the movement. However, the same test applied to the middle phase, ranging from $t = 0.125$ s to $t = 0.45$ s reveals a difference in the motion recognition accuracy between wrist- and sleeve-attached sensors that is statistically significant (i.e. $p < 0.01$). This indicates that the improved accuracy becomes more evident as the human arm moves faster. This trend is also reflected in the cross-fitness distance. This aligns with the findings observed above in relation to the influence of speed on fabric

information content. The difference in discrimination information between the two classes of movement is small during the initial human reaching motion. However, this difference becomes more pronounced as the time window shifts to the middle phase. The improvement in motion recognition accuracy between the wrist-attached and sleeve-attached sensors becomes more apparent during the final phase (ranging from $t = 0.5$ s to $t = 1.25$ s), as shown in Fig. 8d.

## Discussion

This study explores the use of sensorised loose clothing for human motion recognition and prediction. In a wide variety of movement prediction tasks, encompassing different movement frequencies, patterns, and velocities, the performance of motion prediction improves when using loose clothing movement. These tasks include periodic and discrete movements, simple harmonic, linear, curved PTP movements, and circular trajectories, and in movements captured from human participants. Counterintuitively, loose clothing enhances motion prediction performance, shown in higher recognition accuracy

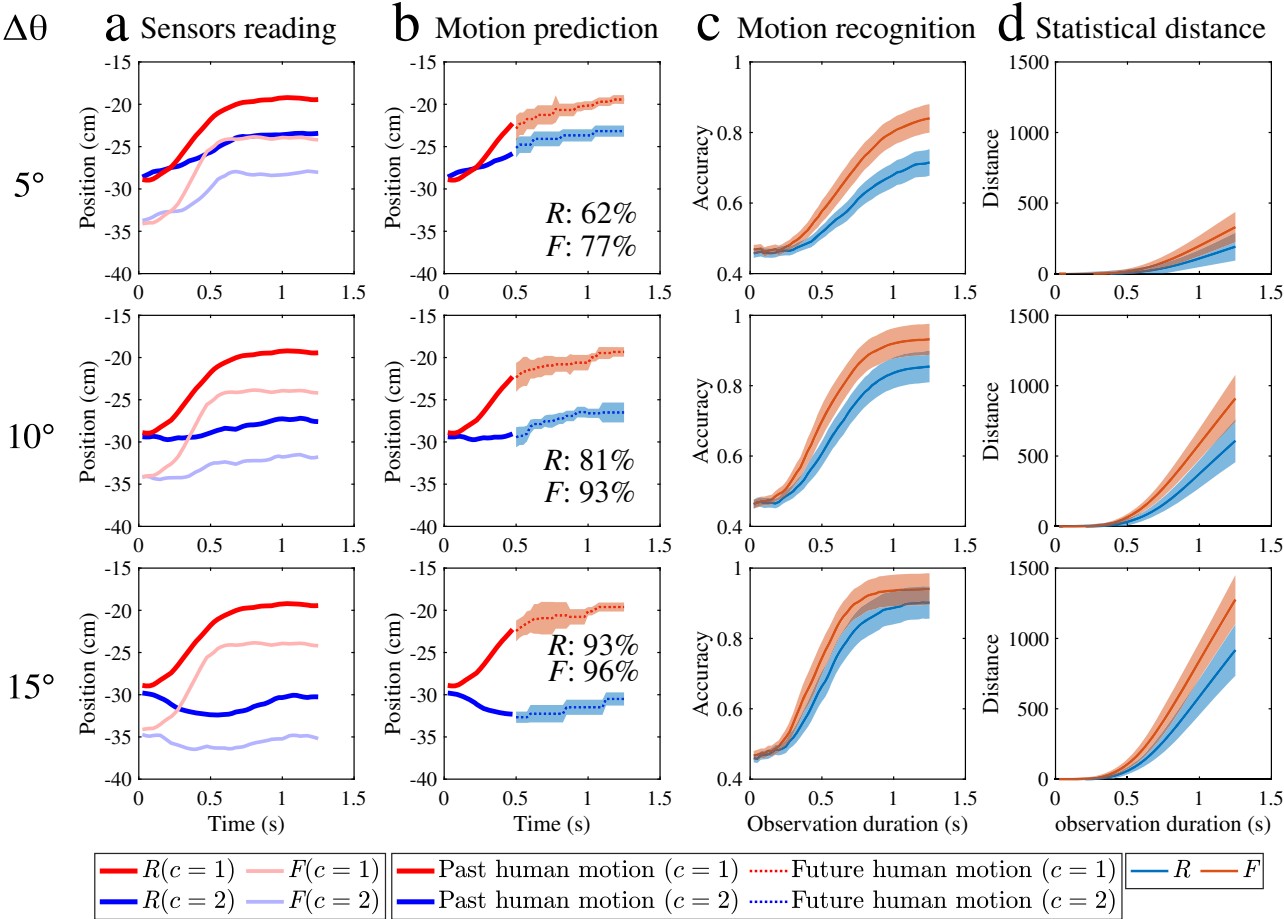

**Fig. 8 | Human reaching is predicted more accurately from loose, sensorised garments. a** The actual vertical position of sensors $R$ and $F$ for reaching two target buttons with different angles. **b** The predicted body motion (mean value dashed line; ±s.d. × 3 is shaded area) given the past body or clothing movement from the initial time up to 0.5 s, and the motion recognition accuracy of both sensors. **c** The motion recognition accuracy and **d** the cross-fitness distance (mean value solid line; ±s.d. is shaded area) of wrist-attached and sleeve-attached sensors between two target buttons with angle differences.

and requiring a shorter duration of past trajectory analysis to achieve satisfactory accuracy. This is especially pronounced in scenarios where the difference between movement classes is small and the movement of the rigid body is fast. These results suggest that the motion of the fabric contains latent information about the movement, beyond what can be gleaned from the motion of the body. This is supported by examining the statistical (i.e. cross-fitness) distance that considers not only the differences between the mean values of movements but also their probability distributions. The statistical distance is higher between movements of clothing than body movement alone, indicating the presence of more discriminatory information, thereby simplifying the task of distinguishing between two movements. The robustness and generalisability of this finding are substantiated through a probabilistic model outlined in Supplementary Note 12, which provides a theoretical framework for explaining the observed phenomenon.

This study presents several intriguing insights, yet it also highlights areas needing further exploration. For example, while the integration of clothing movement appears to significantly enhance motion prediction accuracy, this improvement does not seem to directly correlate with the sensor location on the garments, or the distance from the point of attachment of the fabric to the rigid body. This discrepancy could be attributed to several possible factors, for instance, there may be harmonic or resonance-like effects in how the fabric is excited or deforms that are dependent on the characteristics of the specific motion. It would be interesting to explore this further to determine what might affect this and thereby understand the 'optimal' looseness of clothing. Moreover, certain experiments were conducted in a controlled environment (i.e. airflow velocity, room temperature, and humidity). Airflow velocity was found to play a more critical role in the motion recognition performance of fabric motion. Higher airflow velocities induced significant changes in fabric movement, thereby altering the accuracy of motion recognition using fabric motion. By contrast, room temperature and humidity exhibited no obvious influence on the motion recognition performance of fabric motion. Through these experiments, subjects can adjust the ambient environmental conditions to fully utilise the advantages brought by clothing-mounted sensors.

A limitation of this study is that most of the experiments relied on a non-portable, tethered motion capture system, while many practical applications call for portable sensing solutions such as those based on IMUs. Although we conducted a proof-of-concept validation using IMUs, this remains preliminary in scope and statistical analysis. It should also be acknowledged that the current analysis supporting the applicability of IMUs is constrained by the simplicity of our validation approach. Future work should include more extensive experiments and stronger statistical evaluations to substantiate the transferability of our findings to IMU-based wearable systems.

Nevertheless, the implications of this work span a wide range of domains. From a technological standpoint, the findings of this study

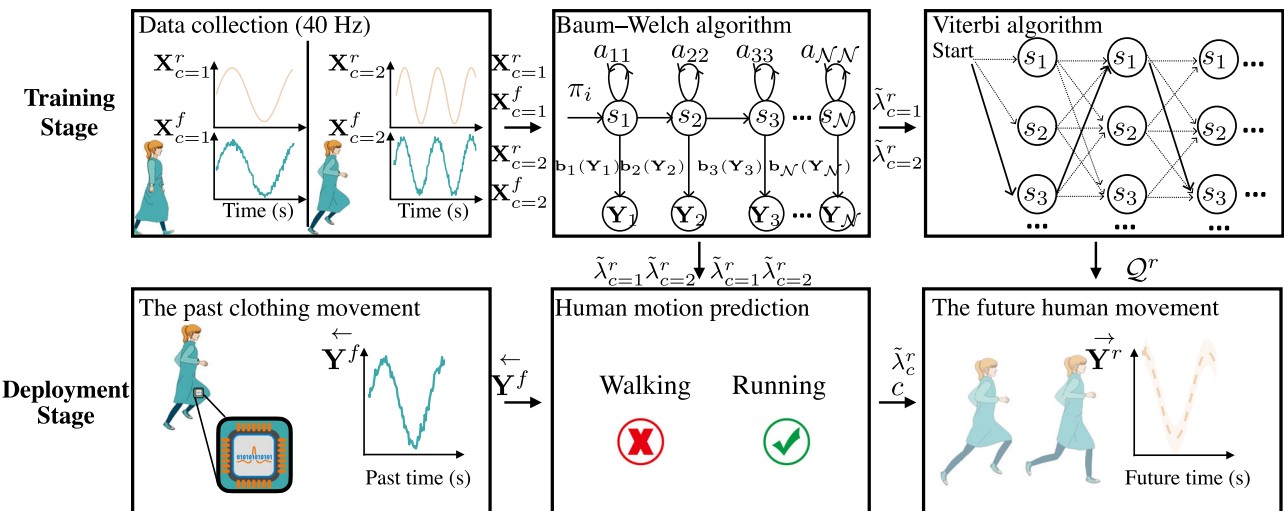

**Fig. 9 | The overview framework for human motion recognition and prediction based on LR-HMM.** The future human movement $\overrightarrow{\mathbf{Y}^r}$ (dashed line and shaded area) is predicted based on the recognition label $c$ of the past clothing movement $\overleftarrow{\mathbf{Y}^f}$ (solid line) and the probability trajectory model $\tilde{\lambda}_c^r$ formulated by body movement using the Baum-Welch algorithm and the Viterbi algorithm.

suggest that sensors should be attached to loose clothing to enhance motion recognition and prediction, instead of using tight-fitting or body-attached sensors as is currently standard practice. For instance, in smart manufacturing environments, where skilled workers collaborate with robotic arms and other machinery, instrumenting their clothing may lead to improved working efficiency (e.g. allowing the robot to detect task-oriented actions and provide assistance more quickly and accurately) and enhanced safety (e.g. predicting the future movement of the worker to avoid collisions with them or their workspace)[49]. In virtual reality games, fabric-based motion recognition and prediction can enable rapid prediction of gaming events (e.g. hand collisions with other users or non-player characters), facilitating pre-rendering of graphics or instantaneous haptic feedback[50,51]. In healthcare settings, predicting future human motion and trajectory could aid in providing real-time interventions, such as detecting and avoiding falls[52], or controlling exoskeletons[53] and prostheses[54,55]. For such applications, short response times are critical. A computational complexity analysis using the method presented in this work is provided in Supplementary Note 13.

More widely, the findings reported here may also have implications in fields such as biology, anthropology, or the social sciences. For instance, by exploiting additional information content, animals may exploit the movement of their own or other animals' loose appendages for evolutionary advantage. It may be speculated, for instance, that exploiting cues in the movement of loose skin, fur or feathers may convey richer information to animals than would otherwise be available, when considering predator-prey relations (e.g. using fur movement to predict the movement of prey), mating behaviour (e.g. enhancing mating displays through flowing plumage), or other interactions. Some limited evidence of this exists, for instance, (i) in spiders that detect prey through vibrations transmitted across their webs, that allow them to determine the size, speed, and position of their prey, even when it is not directly visible[46,47] and (ii) insects that use sensory organs such as sensilla, setae, or receptors to sense the surrounding environment[56]. From a cultural and anthropological perspective, loose, flowing garments (e.g. skirts, tassels, furs) so they are extensively used in modern and traditional costume, especially for performative activities such as dance and religious rituals[42]. The enriched information content of such clothing may explain their popularity in these settings, to enhance the expression of the wearer's through such performances.

## Methods

### Motion prediction problem definition

The problem addressed in this paper is that of predicting future human body movement $\overrightarrow{\mathbf{Y}^r} = [\mathbf{y}_{\nu+1}^r, \mathbf{y}_{\nu+2}^r \dots \mathbf{y}_{\mathcal{V}}^r]$ based on the movement of clothing $\mathbf{Y}^f = [\mathbf{y}_1^f, \mathbf{y}_2^f \dots \mathbf{y}_\nu^f]$ using a statistical model trained on past movements.

The latter is trained on data in which body and clothing-attached sensors are used to simultaneously record movement for $\mathcal{V}$ time steps with $\mathcal{M}$ features as a person performs $\mathcal{K}$ repetitions of a given motion of a predefined class. For example, the sensors may record the multi-dimensional orientations of the human body and clothing when a person is walking (see Fig. 9). Each movement recorded consists of $\mathcal{V}$ sensor readings, i.e. $\mathbf{Y}_k = [\mathbf{y}_1, \mathbf{y}_2 \dots \mathbf{y}_{\mathcal{V}}] \in \mathbb{R}^{\mathcal{M} \times \mathcal{V}}$, and is assumed to belong to one of a finite, discrete set of motions/classes (e.g., walking, running). All prediction tasks are assumed to be binary throughout the paper. The complete set of recordings consist of the body- $\mathbf{X}^r = [\mathbf{Y}_1^r, \mathbf{Y}_2^r, \dots \mathbf{Y}_{\mathcal{K}}^r] \in \mathbb{R}^{\mathcal{M} \times \mathcal{K} \times \mathcal{V}}$ and the clothing-attached sensor readings $\mathbf{X}^f = [\mathbf{Y}_1^f, \mathbf{Y}_2^f, \dots \mathbf{Y}_{\mathcal{K}}^f] \in \mathbb{R}^{\mathcal{M} \times \mathcal{K} \times \mathcal{V}}$. At deployment (i.e. once the model has been trained) it is assumed that no further direct sensing of the body is available, i.e. motion recognition and prediction rely solely on data from fabric-attached sensors (see Fig. 9).

### Data acquisition and pre-processing

**Body and clothing motion acquisition system.** NDI's Aurora Electromagnetic tracking system (NDI Aurora Magnetic Tracking device, NDI, Canada) is used for collecting rigid and fabric motion. Aurora 6 degrees-of-freedom sensors can capture their positional and orientational movements (both are in three dimensions) at a sampling frequency of $f_s$ = 40 Hz. These sensors are lightweight and compact ($0.92 \times 9.4$ mm), thus not influence fabric or clothing movement. At high speeds NDI device occasionally loses track of its sensors resulting in missing data (i.e. gaps) within trajectories. These are filled using piecewise cubic spline interpolation.

**Movements of different frequency.** The scotch yoke is used as the physical mechanism to generate one-dimensional movement with different frequencies. The device consists of a sliding yoke, a rigid rod, two bearing blocks, a rotating disk of radius 10 cm and a DC motor (30: 1, 37$D$ gear-motor, Pololu Corporation, USA) fixed at the fulcrum. The position of the rigid yoke over time is a simple harmonic motion. A piece of fabric ($30 \times 5$ cm strip of woven cotton) was attached at the tip of the yoke. One sensor was attached at the fabric attachment point,

10 cm away from the fulcrum, (i) namely $R_1$. Three sensors were attached to the fabric. Specifically, they were attached along the length of the fabric (ii) 20 cm ($F_2$), (iii) 30 cm ($F_3$) and (iv) 40 cm ($F_4$) from the fulcrum (i.e. at the tip of the fabric). It is beneficial to understand how varying sensor placements on the fabric influence motion prediction accuracy. Figure 2 shows the experimental setup and the scotch yoke is moved at two frequencies. More details can be found in Supplementary Video 1.

The scotch yoke was operated at seven frequencies (i.e. $\omega = \{0.63\pi, 0.75\pi, 0.92\pi, 1.1\pi, 1.2\pi, 1.35\pi, 1.45\pi rad\, s^{-1}\}$) using different pulse width modulation (PWM) values controlled by the Arduino board. With this setup, $\mathcal{K} = 50$ trajectories, each with a randomly assigned starting position and a length of $T = 5$ s, were recorded at each frequency, resulting in $\mathcal{V} = f_s \times T = 40 \times 5 = 200$ data points in each trajectory. The scotch yoke incorporates a pause time of 10 s between each trajectory. All trajectories are aligned temporally so that the positions of sensors rigidly attached are nearly identical at the initial time step ($v = 1, t = 0$) of each trajectory.

**Discrete multidimensional movement patterns.** KUKA's LBR iiwa robot arm (KUKA, Germany) is used to execute linear and curved PTP and circular movement encompassing both linear (SLIN) and curved (SCIRC) motion, using KUKA Sunrise.OS 1.11. The experimental setup is identical to the one used in the preceding section, with the only difference being that the fabric is attached to the end-effector of the robot arm. Figure 4a, b illustrates the experimental setup. Figure 4c shows the three types of predefined trajectories traced by the end-effector of the robot arm. The end effector of the robot arm's moving trajectories in this experiment is described as follows. For linear and curved PTP movement, the first PTP movement is identical. After the first PTP movement is executed, the directions between two classes of movement then completely diverge. For circular movement, the angle formed from the starting position to the centres of the two classes of circular movements is 1°. The starting and ending positions are consistent across all trajectories. Because the initial segments of the two types of movement are nearly identical, this is regarded as a hard prediction task. The moving trajectory and the yaw orientation of the rigid-attached sensor ($R_1$ thick and dark line) and the fabric-attached sensor ($F_4$ thin and light line) are shown in the harder part of Fig. 4c. To assess prediction tasks of differing difficulty, the trajectory (shown in blue in the harder part of Fig. 4c) is rotated around the starting position at various angles (i.e. $\theta = \{10°, 20°, 30°, 40°, 50°, 60°, 70°, 80°, 90°\}$, for more details see Supplementary Video 3). With this setting, the differentiation between two classes of movements becomes more evident as the rotation angles increase between the blue trajectory in the harder part of Fig. 4c and the trajectory at each higher rotating angle. Consequently, the prediction task becomes consistently easier. The easier part of Fig. 4c shows the moving trajectory and the yaw orientation of the rigid-attached sensor ($R_1$ thick and dark line) and the fabric-attached sensor ($F_4$ thin and light line) for each type of trajectory when the prediction task is easier (i.e. $\theta = 40°$, for more details see Supplementary Video 2). The moving patterns and the yaw orientation of sensors reading for all difficulties of prediction task can be referred to Supplementary Note 6. The peak moving velocity (i.e. absolute cartesian velocity) of the end effector of the robot arm, following the previously described trajectories, is repeated at $r° = \{25, 22.5, 20, 17.5, 15\,cm\,s^{-1}\}$. With this set up, $\mathcal{K} = 50$ sequences of length $T = 5$ s are recorded for each rotation angle at each moving velocity ($\mathcal{V} = f_s \times T = 40 \times 5 = 200$ data points). The robot arm operates a pause time of 10 s between each trajectory. Each trajectory is time-aligned before data analysis.

**Human reaching motion.** This study is designed to test the hypothesis that the use of loose, sensorised clothing leads to greater accuracy in movement recognition and prediction. Specifically, the null hypothesis is that there is no significant difference in motion recognition accuracy between the wrist-attached sensor and the sleeve-attached sensor. Conversely, the alternative hypothesis contends that there is a significant difference in motion recognition accuracy between the wrist-attached sensor and the sleeve-attached sensor.

The experimental setup includes five buttons and five LED lights. The white button marks the hand's starting position, while the other four buttons represent different target positions. The distance between each target button and the start button is 45 cm, ensuring uniform travel durations to reach each target. The angle between two adjacent target buttons is 5°, so all target buttons are positioned on an arc centred on the start button (see Supplementary Note 11). To deactivate a target LED light, the participant must press the button of the corresponding colour.

The participants are provided with an information sheet and signed a consent form before visiting the laboratory. Before data collection, the participants' sitting positions are standardised. The distance between the centre of the shoulder joint and the reference point is 70% of the participant's arm length. The sleeve is adjusted to ensure the cuff is aligned flush with the wrist using tape. All data collection is conducted exclusively in an audio-visual isolated room. To familiarise the participant with the movement, a brief video is presented to demonstrate the specific movements to be performed. Upon completion of the data collection, a number of participants will be randomly selected to receive a £25 voucher as a thank you for their participation and contribution.

The participant is asked to remove or roll up the sleeve of any clothing on their arms and to wear an instrumented shirt (100% woven cotton) with one sensor (denoted as $F$, see Supplementary Note 11a) attached to the cuff of the right-side sleeve (with a maximum possible displacement from the wrist of ±10 cm) and another sensor attached near the unciform bone (denoted as $R$, see Supplementary Note 11a). The participant is asked to clench the dominant hand into a fist and use the thumb to fully press the start button, which turns off the corresponding LED light. Subsequently, the subject reaches directly for and presses the button corresponding to the randomly illuminated LED within 2.5 s. The participant's hand then returns to the starting position, awaiting the re-illumination of the LED light. This procedure is repeated 40 times, with each target LED illuminating 10 times per block. The experiment involved a total of 5 blocks. The release times of all buttons are recorded. Supplementary Note 11b is the flow chart of LED lights and the blue target button. An example of data collection during human reaching motion can be found in Supplementary Note 11c. The details of one human reading trajectory can be found in Supplementary Video 5.

A power analysis for repeated measures ANOVA, involving repeated measures, within factors, reveals that for a medium effect size (0.3) and a power of 0.9, the necessary sample size is 22[57]. Accordingly, 22 healthy participants recruited from the local community (14 males, 8 females; mean ± standard deviation; 25.2 ± 3.5 years old; height 174 ± 7.4 cm; mass 74 ± 17 kg; arm length 51 ± 3.5 cm; wrist circumference 16.5 ± 1.7 cm; laterality index 79 ± 17 assessed by Edinburgh handedness questionnaire[58]) took part in this experiment. As a result, an equal number of samples (50 for each target button) are recorded. The start time index of each trajectory is determined when the change in the horizontal dimension of the wrist-attached sensor's movement exceeds 10 mm. Each trajectory occurring within this designated time interval originates from its respective start time index. The duration of each trajectory is extended to 1.25 s per trajectory ($\mathcal{V} = f_s \times T = 40 \times 1.25 = 50$ data points) through linear interpolation.

**Data standardisation.** In the physical mechanism-based experiments outlined, the initial conditions for both rigid and fabric motion are

standardised by adjusting the orientation of sensors attached to rigid and fabric structures for one movement class to a unified baseline of 0° across all axes at the first time step. This adjustment employs a consistent value that is also applied to the other class of rigid and fabric movement. Standardising the starting points in this way ensures the preservation of relative differences between subsequent data points within each data series, thus facilitating comparisons between the two distinct classes of movement and streamlining computational processes.

**Time alignment of each trajectory.** To ensure consistency among all trajectories at each time step, it is essential to time-align each trajectory. This process involves synchronising the starting points of the movements across all trajectories. The position of sensor $R_1$ is used to find the starting time indices of each trajectory. Specifically, the starting index is determined by identifying when the difference between the positions of two adjacent time indices of the horizontal dimension of sensor $R_1$ exceeds certain thresholds during a reasonable time interval. In the case of experiments of different frequency movements, discrete multidimensional movement patterns and human reaching motion, these thresholds are set at 1 mm, 0.5 mm, and 10 mm, respectively. The starting time indices of sensor $F_2$, sensor $F_3$ and sensor $F_4$ movements are consistent with the starting time indices of sensor $R_1$ movement. These values are selected to be as small as possible, while still ensuring successful time alignment.

**Baseline correction for hand-made Scotch yoke movement.** The motion of the hand-made Scotch yoke may vary across different experiments. (i) The position of the Scotch yoke shifted after submitting the first manuscript, and its vertical orientation may have differed. This altered the distance between the yoke and the rotation disk, significantly affecting the friction on the Scotch yoke and, in turn, its movement. (ii) The experiments involve different environments (i.e. different airflow velocities, room temperatures and humidity levels) requiring long-time operation of the hand-made Scotch yoke. This could also change the distance between the yoke and the rotation disk during this long-time running, thereby influencing the movement of the Scotch yoke. Therefore, baseline correction on the three-dimensional orientation movement of the rigid-attached sensor to standardise the initial values (i.e. 0°) across repeated trials is performed. This addresses the issue of long-term instability in hand-made mechanics. The rigid motion collected by the robot arm does not have this issue. Therefore, this step is only applied to experiments involving the hand-made Scotch yoke.

## Data analysis
**Methods of motion prediction overview.** Figure 9 illustrates the framework of the proposed method. In the training stage, given the observations (i.e. body and clothing-attached sensor readings) of two classes of movements $\mathbf{X}^f_{c=1}$, $\mathbf{X}^f_{c=2}$, $\mathbf{X}^r_{c=1}$ and $\mathbf{X}^r_{c=2}$, the LR-HMMs parameters $\lambda^f_{c=1}$, $\lambda^f_{c=2}$, $\lambda^r_{c=1}$ and $\lambda^r_{c=2}$ are set up and Baum-Welch algorithm is used to estimate parameters $\widetilde{\lambda}^f_{c=1}$, $\widetilde{\lambda}^f_{c=2}$, $\widetilde{\lambda}^r_{c=1}$ and $\widetilde{\lambda}^r_{c=2}$, the most likely state sequence $Q^r$ is estimated using the Viterbi algorithm to formulate the probabilistic trajectory model of the human body movement.

In the deployment stage, the past clothing movement $\mathbf{Y}^f$ (solid line) is then used to recognise the movement (i.e. $c = 1$ or $c = 2$ is determined) using the classification rule. The three-dimensional Euler angles of each sensor's readings are used for recognition (i.e. $\mathcal{M} = 3$). With this recognition, the probabilistic trajectory model of the human movement is used to predict the future human movement $\overrightarrow{\mathbf{Y}^r}$ (dashed line and shaded area).

**Left-to-right hidden Markov model parameter set up.** Consider an $\mathcal{N}$-state Markov chain where the individual states are denoted as $S = \{s_1, s_2, \ldots, s_{\mathcal{N}}\}$. The sequence of states that follows the chronological order of the sensor readings is defined as the most likely state sequence, denoted as $Q = \{q_1, q_2, \ldots, q_{\mathcal{V}}\}$. The most likely state at time step $v$ is $q_v$. The estimated value in the future trajectory is in the continuous measurement space[59].

The transitions between these individual states are defined by the transition probability matrix $\mathbf{A} = \{a_{ij}\}$, whose elements are

$$a_{ij} = P(q_v = s_j | q_{v-1} = s_i),\ 1 \le i, j \le \mathcal{N}. \tag{1}$$

where $a_{ij}$ represents the probability of transitioning to state $s_j$ at time step $v$, conditional on being in state $s_i$ at time step $v-1$. As the sensor reading progresses with each time step, the probability of transitioning to state $s_j$ at time $v$, given that it was in state $s_i$ at time step $v-1$, is much higher than remaining in state $s_i$ at time step $v$. (i.e. $a_{ij} >> a_{ii}$).

The emission probability, denoted as $\mathbf{B}$, is defined as follows: $\mathbf{b}_i(\mathbf{Y}_v)$ represents the probability of an individual state $s_i$ given sensor readings $\mathbf{Y}_v$ at time step $v$. It is defined by the Gaussian probability density function:

$$\mathbf{b}_i(\mathbf{Y}_v) = P(\mathbf{Y}_v | q_v = s_i) = N(\mu_i, \Sigma_i),\ 1 \le i \le \mathcal{N}. \tag{2}$$

The initial (i.e. $v = 1$) state distribution is

$$\pi_i = P(q_1 = s_i),\ 1 \le i \le \mathcal{N}. \tag{3}$$

The number of individual states is set to be equal to the number of time steps in each observation trajectory (i.e. $\mathcal{N} = \mathcal{V}$). This implies that each state represents a specific time step in observation. This choice is designed to maximise the performance of the prediction. This type of hidden Markov model (HMM) is also known as a LR-HMM[60]. A compact notation of the LR-HMM parameters is $\lambda = \{\pi, \mathbf{A}, \mathbf{B}\}$.

**left-to-right hidden Markov model parameter estimation.** The Baum-Welch algorithm is used to estimate the LR-HMM parameters $\widetilde{\lambda}^r_{c=1}$, $\widetilde{\lambda}^r_{c=2}$, $\widetilde{\lambda}^f_{c=1}$ and $\widetilde{\lambda}^f_{c=2}$, given sensor readings $\mathbf{X}^r_{c=1}$, $\mathbf{X}^r_{c=2}$, $\mathbf{X}^f_{c=1}$, $\mathbf{X}^f_{c=2}$[59].
- The initial values of LR-HMM parameters are chosen randomly from the sensor reading.
- E-step: The forward variable is donated as:

$$\alpha_v(i) = P(\mathbf{Y}_1 \mathbf{Y}_2 \ldots \mathbf{Y}_v, q_v = s_i | \lambda), \tag{4}$$

and the backward variable is donated as:

$$\beta_v(i) = P(\mathbf{Y}_{\mathcal{V}} \mathbf{Y}_{\mathcal{V}-1} \ldots \mathbf{Y}_v, q_v = s_i | \lambda). \tag{5}$$

$\gamma_v(i)$ is defined as the probability of being in state $s_i$ at time step $v$, given the observation $\mathbf{X}$ and model $\lambda$. It can be shown

$$\gamma_v(i) = P(q_v = s_i | \mathbf{X}, \lambda) = \frac{\alpha_v(i)\beta_v(i)}{\sum_{j=1}^{\mathcal{N}} \alpha_v(j)\beta_v(j)}, \tag{6}$$

where

$$\alpha_v(i) = \mathbf{b}_i(\mathbf{Y}_v) \sum_{j=1}^{\mathcal{N}} \alpha_{v-1}(j)a_{ji}, \tag{7}$$

and

$$\beta_v(i) = \sum_{j=1}^{\mathcal{N}} \beta_{v+1}(j)a_{ij}\mathbf{b}_j(\mathbf{Y}_{v+1}). \tag{8}$$

$\xi_v(i,j)$ is defined as the probability of being in state $s_i$ at time step $v$, and state $s_j$ at time step $v+1$, given the LR-HMM parameter $\lambda$ and observation $\mathbf{X}$:

$$\xi_v(i,j) = P(q_v = s_i, q_{v+1} = s_j | \mathbf{X}, \lambda) = \frac{\alpha_v(i) a_{ij} \mathbf{b}_j(\mathbf{Y}_{v+1}) \beta_{v+1}(j)}{\sum_{i=1}^{\mathcal{N}} \alpha_v(i) \beta_v(i)}. \quad (9)$$

$\gamma_v^{(\mathcal{K})}(i)$ and $\xi_v^{(\mathcal{K})}(i,j)$ for each observation sequence $\mathcal{K}$ can be computed repetitively using eqs. (6)(9). The new $\widetilde{\lambda}$ can be updated as below[61].

- M-step: update LR-HMM parameter:

$$\pi^{\sim}_i = \frac{\sum_{k=1}^{\mathcal{K}} \gamma_{v=1}^{(\mathcal{K})}(i)}{\mathcal{K}}, \quad (10)$$

$$\mu^{\sim}_i = \frac{\sum_{k=1}^{\mathcal{K}} \sum_{v=1}^{\mathcal{V}} \gamma_v^{(\mathcal{K})}(i) \times \mathbf{Y}_v^k}{\sum_{k=1}^{\mathcal{K}} \sum_{v=1}^{\mathcal{V}} \gamma_v^{(\mathcal{K})}(i)}, \quad (11)$$

$$\sigma^{\sim 2}_i = \frac{\sum_{k=1}^{\mathcal{K}} \sum_{v=1}^{\mathcal{V}} \gamma_v^{(\mathcal{K})}(i) \times (\mathbf{Y}_v^k - \mu^{\sim}_i)^2}{\sum_{k=1}^{\mathcal{K}} \sum_{v=1}^{\mathcal{V}} \gamma_v^{(\mathcal{K})}(i)}. \quad (12)$$

where $\mathbf{Y}_v^k$ is the sensor reading at time $t$ in the $k$th trajectory.

- The E-step and M-step are iterated until LR-HMM parameter $\widetilde{\lambda}$ converges.

**Motion recognition.** The classification rule introduced in ref. 62 is used to make a recognition. This compares likelihoods based on two LR-HMM parameters and the past movement. i.e.

$$c = \underset{c}{\mathrm{argmax}}\, \log P(\overleftarrow{\mathbf{Y}^f} | \widetilde{\lambda}_c^f),\ c \in \{1,2\}$$
$$c = \underset{c}{\mathrm{argmax}}\, \log P(\overleftarrow{\mathbf{Y}^r} | \widetilde{\lambda}_c^r),\ c \in \{1,2\} \quad (13)$$

The forward algorithm is used to compute the likelihoods of the past movement given LR-HMM parameters. It is used to compute the probability of the past trajectory (i.e. the likelihood) given the LR-HMMs $\lambda_c$[59]. The steps of the forward algorithm are:

- Initialisation: the forward probability of the past observation at the first time step (i.e. $v = 1$), is

$$\alpha_1(i) = \pi_i \mathbf{b}_i(\mathbf{Y}_1). \quad (14)$$

- Induction: the recursive formula for the forward probability of the past observations at time step $v$ is (7).
- Termination: summing the forward probabilities $\alpha_v$ of all possible states at the time step $v$

$$P(\mathbf{Y}_1, \mathbf{Y}_2, \ldots, \mathbf{Y}_v | \lambda_c) = \sum_{i=1}^{\mathcal{N}} \alpha_v(i). \quad (15)$$

49 trajectories are randomly chosen from each class of movement of each sensor used to train the separate LR-HMMs $\widetilde{\lambda}_{c=1}^r$, $\widetilde{\lambda}_{c=2}^r$, $\widetilde{\lambda}_{c=1}^f$ and $\widetilde{\lambda}_{c=2}^f$. The remaining trajectories are reserved for testing purposes. Motion recognition is made using Eq. (13). The above-described process is repeated 100 times. The motion recognition accuracy is calculated as

$$\text{Motion recognition accuracy} = \frac{\text{number of correct recognition}}{\text{total number of recognitions}(100 \times 2 = 200)} \quad (16)$$

**Motion prediction.** In this method, each individual state is set to correspond to a specific time step in the observation trajectory. However, it is not explicitly determined which individual state in the estimated LR-HMM corresponds to which time step in the observation trajectory. Formulating the probabilistic trajectory model requires finding which individual state corresponds to each time step of the observation trajectory. Therefore, the most likely state sequence $Q^r$ needs to be solved using the Viterbi algorithm, as described in ref. 63. After that, the probability density function of each individual state given the rigid-attached sensor's reading $\mathbf{Y}_v^r$ at each time step $v$ (i.e. emission probability $\mathbf{B}$) and the most likely state sequence $Q^r$ can be used to form the probabilistic trajectory model (i.e. $\mathbf{b}_{q_1}(\mathbf{Y}_1^r), \mathbf{b}_{q_2}(\mathbf{Y}_2^r), \mathbf{b}_{q_3}(\mathbf{Y}_3^r) \ldots \mathbf{b}_{q_{\mathcal{V}}}(\mathbf{Y}_{\mathcal{V}}^r)$). The most likely state sequence is denoted as $Q = \{q_1, q_2, q_3 \ldots q_{\mathcal{V}}\}$. This problem can be described as below

$$\delta_t = \max_{q_1, q_2, q_3 \ldots q_v} P(q_1 q_2 q_3 \ldots q_v = i, \mathbf{Y}_1 \mathbf{Y}_2 \ldots \mathbf{Y}_v | \mathbf{X}, \lambda). \quad (17)$$

$\delta_v(i)$ is defined as the highest probability along a single path at time step $v$.

$S_v^*$ is estimated using the following steps.

- Initialisation:

$$\delta_1(i) = \pi_i \mathbf{b}_i(\mathbf{Y}_1).$$
$$\Psi_1(i) = 0. \quad (18)$$

- Recursion:

$$\delta_v(i) = \max_i(\delta_{v-1}(i) + a_{ij}) + \mathbf{b}_i(\mathbf{Y}_v).$$
$$\Psi_v(i) = \mathrm{argmax}_i(\delta_{v-1}(i) + a_{ij}). \quad (19)$$

- Termination:

$$P^* = \max(\delta_{\mathcal{V}}(i)).$$
$$q^* = \mathrm{argmax}(\delta_{\mathcal{V}}(i)). \quad (20)$$

- Path (state sequence) backtracking:

$$q_v^* = \Psi_{v+1}(q_{v+1}^*),\ v = \mathcal{V} - 1, \mathcal{V} - 2, \ldots, 1. \quad (21)$$

The emission probability and the most likely state sequence $Q$ can be used to form the probabilistic trajectory model (i.e., $\mathbf{b}_{q_1}(\mathbf{Y}_{v=1}^r), \mathbf{b}_{q_2}(\mathbf{Y}_{v=2}^r), \mathbf{b}_{q_3}(\mathbf{Y}_{v=3}^r) \ldots \mathbf{b}_{q_{\mathcal{V}}}(\mathbf{Y}_{\mathcal{V}}^r)$).

**Statistical Distance between Left-to-right hidden Markov models.** The statistical distance between LR-HMMs given two categories of movements is computed as a measure of discrimination information between two classes of LR-HMMs[64]. The cross-fitness distance of rigid movements $D^r$ and fabric movements $D^f$ is computed as proposed in ref. 65 according to

$$D^r = \log P(\overleftarrow{\mathbf{Y}_{c=1}^r} | \tilde{\lambda}_{c=1}^r) + \log P(\overleftarrow{\mathbf{Y}_{c=2}^r} | \tilde{\lambda}_{c=2}^r) - \log P(\overleftarrow{\mathbf{Y}_{c=1}^r} | \tilde{\lambda}_{c=2}^r) - \log P(\overleftarrow{\mathbf{Y}_{c=2}^r} | \tilde{\lambda}_{c=1}^r),$$
$$D^f = \log P(\overleftarrow{\mathbf{Y}_{c=1}^f} | \tilde{\lambda}_{c=1}^f) + \log P(\overleftarrow{\mathbf{Y}_{c=2}^f} | \tilde{\lambda}_{c=2}^f) - \log P(\overleftarrow{\mathbf{Y}_{c=1}^f} | \tilde{\lambda}_{c=2}^f) - \log P(\overleftarrow{\mathbf{Y}_{c=2}^f} | \tilde{\lambda}_{c=1}^f). \quad (22)$$

The forward algorithm can be used to compute each term in Eq. (22).

## Ethics statement

The human reaching experiment was conducted with the ethical approval of King's College London, UK (MRPP-23/24-40031).

## Reporting summary

Further information on research design is available in the Nature Portfolio Reporting Summary linked to this article.

## Data availability

All data generated in this study have been deposited in the Figshare repository and are accessible at https://figshare.com/s/4b69b5734100c9762fed.

## Code availability

The custom MATLAB scripts used for solving motion recognition and prediction tasks are available in the Figshare repository at https://figshare.com/s/4b69b5734100c9762fed.

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

## Acknowledgements

We extend our sincere gratitude to all participants involved in the human reaching experiment for their time and patience. We are also grateful to Michael Gomez and Zoran Cvetkovic for their valuable suggestions on the manuscript prior to its submission. This work was partially supported by King's College London, the China Scholarship Council, and the Engineering and Physical Sciences Research Council (EP/M507222/1).

## Author contributions

T.S. and M.H. proposed the idea and designed the experiments. T.S. and S.M. implemented the hardware setup and collected data. T.S. formulated left-to-right hidden Markov models and obtained the experimental results. T.S., I.D.G., and M.H. analysed the results. T.S. wrote the original manuscript, and M.H. reviewed and edited it.

## Competing interests

The authors declare no competing interests.
