## [Transparent Peer Review file · Nature Communications]

Human Motion Recognition and Prediction using Loose Cloth

Corresponding Author: Dr Tianchen Shen

Version 0:

Reviewer comments:

Reviewer #1

(Remarks to the Author)

The study presents interesting and innovative contributions, particularly regarding the use of loose garments with sensors to improve human movement recognition and prediction, challenging the traditional idea that sensors must be rigidly attached to the body. However, there are several methodological and validation issues that need significant improvements before the study can be considered for publication.

- This study is based solely on empirical evidence, without a clear theoretical explanation or justification. Without a clear theoretical explanation, the results can seem incomplete or not robust because it's unclear why specific outcomes occur. This reduces the replicability or generalizability of the results. Generally, high-quality scientific journals do require a thorough discussion that not only presents results but also attempts to explain the phenomenon or offer hypotheses, at least preliminary, that can be tested in future studies. For example, a critical step would be to verify if the superior performance of fabric sensors is due to fabric noise being interpreted as useful signals. To do this, the authors could (a) conduct experiments where the fabric is fixed to limit oscillations while keeping body movement constant. If accuracy decreases significantly, it could indicate that fabric oscillations were providing additional (possibly noisy) information. (b) introduce artificial noise to the data to see if the model's performance decreases. If the model continues to perform well despite added noise, it suggests the system is robust and not dependent on random artifacts.

In addition:

- the authors could analyze the noise present in the data collected from fabric-mounted sensors. Tools like Fourier transforms or noise filtering techniques (e.g., Kalman filters) could help distinguish fabric oscillations caused by body movement from unrelated oscillations.
- In order to avoid an intrinsic bias of the LR-HMM model, it could be compared with other advanced models, such as Recurrent Neural Networks (RNNs) or Long Short-Term Memory (LSTM) networks to confirm that the improvement is not due to bias in the HMM model.
- The authors could conduct a correlation analysis to see if fabric oscillations are tightly correlated with body movements. Measure how synchronized fabric movement (recorded by the fabric-mounted sensors) is with actual body movement. If there's a strong correlation, it would indicate that the fabric is amplifying body movement in a useful way. If correlation is low, it's possible that the improvement is due to noise artifacts.
- Using different fabrics with varying physical properties (e.g., rigid vs. flexible fabrics) could help better understand the role fabric plays in improving performance. If sensors on more rigid fabrics do not show improvement compared to flexible ones, it could confirm that flexible fabric movement is useful for motion recognition.

Finally,

- the experiments were conducted in extremely controlled environments (e.g., with robotic devices or a scotch yoke), which may not reflect the complexities of real-world situations. The authors themselves acknowledge that external environmental factors (such as wind, temperature, and humidity) were not considered. It would be helpful to include experiments or simulations that consider more realistic contexts, perhaps in outdoor environments or with varying environmental factors. This could help demonstrating that the system is applicable beyond the lab.
- There's a human-reaching experiment in which the two types of sensors, mounted on the body and on fabric, were compared. Also in this case, although the experiment outcomes are very promising results, the tested reaching task is relatively simple and predictable. More complex movements, such as 3D or irregular movements, were not considered. It would be necessary to test the system on more complex and dynamic movements to verify whether fabric-mounted sensors maintain their superiority. Again, the experiment was conducted in a highly controlled environment, with known position

targets and a fixed setup. In real-world contexts, where external interferences (e.g., wind moving the fabric) could occur, the results might change. Conducting experiments in less controlled environments would help to assess how the system behaves with additional environmental variables. Although the study includes 22 participants, it would be advisable to test the system on a larger and more diverse sample, including different types of movement, execution styles, and physical characteristics of the participants. More natural and less standardized movements could present challenges for the system, especially if the fabric behaves unpredictably.

- The use of LR-HMMs, combined with the forward-backward algorithm and Viterbi algorithm, can be computationally expensive, especially when increasing the number of states. It is unclear whether the proposed method is scalable to large datasets or applicable in real-time, especially in applications like virtual reality or human-robot collaboration, as suggested. The authors should provide an analysis of computational complexity and discuss how the system would perform on larger datasets or in real-time applications.

(Remarks on code availability)

Reviewer #2

(Remarks to the Author)

This paper focuses on predicting human motion patterns from loose clothing. It is a topic of interest to the researchers in the related areas. Specifically, this paper proposes an approach to enhance human motion recognition and prediction through sensors embedded in loose clothing, demonstrating that the fabric motion within loose clothing provides additional potential information, thereby improving the accuracy of motion prediction. However, using Left-to-Right Hidden Markov models for motion prediction is already a relatively mature research field. While studying the impact of loose clothing on motion prediction offers some novelty, it lacks profound theoretical or technical breakthroughs. Therefore, the paper needs very significant improvement before acceptance for publication. Detailed comments are as follows:

1. Compared to the Hidden Markov Models reported for motion monitoring, please provide innovations in the algorithm. Attempt to review the literature related to motion recognition and prediction, systematically comparing the advantages and disadvantages of sensors in tight-fitting clothing or directly on the body.
2. As the author mentioned, exploring the impact of real-world environmental conditions on motion prediction for loose clothing is particularly important. Certain conditions like wind speed, humidity, and temperature can be simulated in a lab setting, and incorporating some of these findings into this paper would enhance the study's realism and applicability.
3. Compared to Aurora 6 degrees-of-freedom sensors, accelerators/inertial measurement units are more widely used. Please provide some experimental data to demonstrate that they still support the conclusions of this paper.
4. Beyond the button press task, running scenarios can be added to further enhance the adaptability of the prediction model.
5. The motion of clothing contains additional information useful for analyzing human movement. By conducting experiments/simulations/theoretical modeling, the true relationship between the motion information of sensor-attached loose clothing (rather than a simple piece of fabric) and human movement can be quantified, further highlighting the importance of such additional information in motion prediction.

(Remarks on code availability)

Version 1:

Reviewer comments:

Reviewer #1

(Remarks to the Author)

The authors' responses present a number of clear strengths. First, they provide a well-articulated theoretical explanation for the observed benefits of fabric-based sensing. The probabilistic model introduced in the supplementary material, grounded in the concept of Kullback-Leibler divergence, offers a solid and intuitive rationale for why motion recognition improves when using loosely attached sensors. This modeling work is both insightful and well-aligned with the experimental results. In addition, the authors undertook significant new experimental work to test their approach under variable environmental conditions. The simulations of airflow, humidity, and temperature are particularly valuable in demonstrating the robustness of the method in more realistic settings. These tests confirm that, under typical conditions, the fabric sensors perform reliably, and in some cases, better than rigid sensors. Only at higher wind speeds does their performance start to decline. Furthermore, the authors show that their findings are not tied to a single statistical approach by implementing and tuning an LSTM model, which, although less effective than the LR-HMM, still supports the main claims. This strengthens the argument that the improvement is due to the nature of the sensing rather than to the specifics of the model used.

Nonetheless, there are still some important issues that remain partially addressed. The comparison of learning models, while welcome, is limited to just one alternative (LSTM) and is confined to a single experimental scenario. A broader analysis involving additional models, such as GRUs or temporal convolutional networks, and across different types of movement would provide stronger evidence of generalizability. More importantly, the authors did not provide any validation using standard IMUs. While they justify their focus on textile-embedded sensors as a future direction, IMUs remain the current standard in many fields, and even a minimal empirical comparison would have strengthened the practical credibility

of their claims.

Finally, although the environmental experiments are noteworthy, the lack of formal statistical tests (e.g., ANOVA or similar) somewhat limits the strength of the conclusions regarding robustness. Including such analysis would help quantify how much environmental variation truly impacts recognition performance.

(Remarks on code availability)

Reviewer #2

(Remarks to the Author)

The authors have thoroughly revised the manuscript to address the concerns raised. The quality of the manuscript has significantly improved and is now suitable for publication consideration.

(Remarks on code availability)

Version 2:

Reviewer comments:

Reviewer #1

(Remarks to the Author)

The revised manuscript has been significantly improved. The addition of TNN and TCN, the IMU proof-of-concept, and the expanded statistical analyses and experiments clearly strengthen the paper. The manuscript is now clear and well-structured.

Before acceptance, I'd only ask for two minor clarifications. Please make it explicit that the IMU experiment is a proof-of-concept and that a more systematic validation is left for future work. It would help to include a short note in the Discussion acknowledging the limits of the current statistical analysis and why more complex methods weren't applied here.

(Remarks on code availability)

Reply to the review report on

Can your Shirt Predict your Movement? Improving Motion Analysis using Loose Clothing

by Tianchen Shen, Sacha Morris, Irene Di Giulio and Matthew Howard

May 31, 2025

Thank you for giving us the opportunity to submit a revised draft of the manuscript “Can your Shirt Predict your Movement? Improving Motion Analysis using Loose Clothing” for publication in *Nature Communications*.

We appreciate the thoughtful consideration you have given to our work. We have carefully checked all the reviewers’ comments and made corresponding changes in the manuscript.

In this response letter, the reviewers’ original comments are highlighted with a red background, our replies appear in blue background, and the amendments made to the manuscript are enclosed in quotation marks and distinguished by a green background. In line with the reviewers’ comments, we have also created Supplementary Information to present the results of additional new experiments.

For ease of reference, we have assigned a unique number to each comment in the form CX.Y, where X refers to the reviewer number and Y to the comment of that reviewer. Below, we provide detailed responses to the reviewers’ comments.

Reviewer 1

C1.1. General Comment

This paper focuses on predicting human motion patterns from loose clothing. It is a topic of interest to the researchers in the related areas. Specifically, this paper proposes an approach to enhance human motion recognition and prediction through sensors embedded in loose clothing, demonstrating that the fabric motion within loose clothing provides additional potential information, thereby improving the accuracy of motion prediction. However, using Left-to-Right Hidden Markov models for motion prediction is already a relatively mature research field. While studying the impact of loose clothing on motion prediction offers some novelty, it lacks profound theoretical or technical breakthroughs. Therefore, the paper needs very significant improvement before acceptance for publication. Detailed comments are as follows:

Our response

Thank you for confirming the view that this work is of interest to researchers across fields.

Our aim in this paper is to investigate how the physical effects of loose clothing affect motion recognition and prediction tasks in general, regardless of the particular statistical machine learning technique used. To this end, hidden Markov models are used precisely because they are a well-established technique for studying time-series data. This gives us confidence that our findings may be attributed to the effect of the loose clothing rather than due to any (potentially as-yet poorly understood) effect of newer, more experimental machine learning techniques.

Moreover, the probabilistic framing of hidden Markov models (as opposed to, for example, non-probabilistic techniques, such as neural networks) lends itself to better understanding of the information content of the fabric motion. To this end, a probabilistic model explaining the observed phenomena is presented in Supplementary Information S2 (line 8-60). This framework explains improved responsiveness and accuracy with fabric sensing from the increased statistical distance between movements recorded.

Specifically, the model shows that statistical distance measures such as Kullback-Leibler divergence imply that stochastic fabric movements lead to greater discriminative ability. The predictions of this model support the finding of this paper, where it is evident that sensors loosely attached to fabric yield greater accuracy than rigidly attached ones, especially when the prediction task is challenging.

Changes:

We have introduced a probabilistic model explaining the observed phenomena. It is presented in Supplementary Information S2 (line 8-60).

C1.2. Comments to the authors

1. Compared to the Hidden Markov Models reported for motion monitoring, please provide innovations in the algorithm.

Our response

Thank you for your comment. We would like to clarify that our study does not claim to introduce novel algorithmic modifications to the core Left-to-Right Hidden Markov model (LR-HMM) structure, which is already well-regarded for its robustness and effectiveness in modelling sequential data such as human motion. Our focus in this study is on whether the effects of loose clothing may be beneficial with regards to motion analysis, *irrespective of the statistical modelling technique*, and if so, *how they may be exploited* (here, using Left-to-Right Hidden Markov models (LR-HMMs) as an example).

The choice of LR-HMMs is largely a practical one, since (i) LR-HMMs are well-suited for modelling time series data across various applications [1] due to their ease of construction, manipulation, and the existence of optimal algorithms for training and scoring, such as the forward algorithm and the Viterbi algorithm [2], (ii) LR-HMM can solve motion prediction and recognition problems using a single trained model, and (iii) compared to other methods (*e.g.*, recurrent neural network (RNN)), the Markov model is a probabilistic framework that aids understanding this physical phenomenon. For example, the statistical distance between LR-HMMs given two categories of movements is computed as a measure of discrimination information between two classes of LR-HMM. Nevertheless, in response to this comment, we have repeated the analysis using other

statistical modelling techniques (e.g., Long Short-Term Memory) with similar results to those reported for LR-HMMs (please refer to C2.6 for details). Text has been added to the manuscript noting this (see Supplementary Information S3, line 61-84), however, for reasons of clarity and brevity, the full analysis of these results is not reported.

C1.3. Comments to the authors

1.[continued] Attempt to review the literature related to motion recognition and prediction, systematically comparing the advantages and disadvantages of sensors in tight-fitting clothing or directly on the body.

Our response

Thanks for your suggestion. We use a table to systematically summarise the advantages, disadvantages and applicable users for different types of motion capture systems (see Table R1). This table highlights the advantages of clothing-attached sensors in terms of factors such as comfort and ease of use.

Changes:

We added Table R1 in Supplementary Information S1.

C1.4. Comments to the authors

2. As the author mentioned, exploring the impact of real-world environmental conditions on motion prediction for loose clothing is particularly important. Certain conditions like wind speed, humidity, and temperature can be simulated in a lab setting, and incorporating some of these findings into this paper would enhance the study's realism and applicability.

Our response

Thank you for your comment. We agree that investigating the impact of real-world environmental conditions on motion prediction for loose-fitting clothing is valuable. To investigate this, we conducted controlled repetitions of the Scotch yoke experimental configuration under varying environmental parameters within a laboratory environment. A multifunctional climate control unit (3-in-1 air cooler, purifier and humidifier, Sealey

Types of Motion Capture System	Advantages	Disadvantages	Applicable Users
Marker-based Optical	High accuracy and sampling frequency, real-time processing	Uncomfortable, complex setup, and small capture volume within a laboratory environment	Researchers, biomechanical analysts, film/game industry, rehabilitation specialists
Markerless Optical	User-friendly, allows movement in a natural environment	Constrained by environmental requirements, lower precision and real-time capabilities	Game developers, Augmented Reality and Virtual Reality applications, fitness tracking, human-computer interaction researchers
Wearable Sensor-based	Measures movements without an external reference, greater portability	Uncomfortable, complex setup	Sports scientists, physical rehabilitation, exoskeleton and prosthetics researchers
Magnetic	High accuracy, immune to line-of-sight issues	Sensitive to magnetic interference from metal objects, small capture range, cannot track high-speed objects	Medical applications (surgery), human-computer interaction, aerospace and virtual reality simulations
Mechanical	High accuracy and efficient	Discomfort, large size, environmental constraints	Robotics, biomechanics research, exoskeleton control and prosthetics development
Sensorised Garments (Our approach)	Comfortable (if loose-fitting), easy to use, unobtrusive, portable, extended use, allows movement in a natural environment	Limited exploration of utilising clothing movement for human motion analysis	Everyday users, people who demand high levels of wearing comfort (e.g., the elderly), healthcare and rehabilitation assessments

Table R1: Comparison of different types of motion capture systems.

UK) was employed to systematically control three key variables: airflow velocity, room temperature and humidity. Figure R1 depicts the experimental setup, utilising a wind speed meter to measure airflow velocity and a digital hygrometer to monitor ambient temperature and humidity levels.

We found that the movement of the Scotch yoke differs from that in the first submitted manuscript, which can be attributed to the following reasons: (i) The position of the Scotch yoke shifted after submitting the first manuscript, and its vertical orientation may have differed. This altered the distance between the yoke and the rotation disk, significantly affecting the friction on the Scotch yoke and, in turn, its movement. (ii) The added experiments involve different environments (*i.e.*, different airflow velocities, room temperatures and humidity levels) requiring long-time operation of the hand-made Scotch yoke. This could also change the distance between the yoke and the rotation disk during this long-time running, thereby influencing the movement of the Scotch yoke. Therefore, we performed baseline correction on the three-dimensional orientation movement of the rigid-attached sensor to standardise the initial values (*i.e.*, 0 degree) across repeated trials. This step was also applied to the original data, and adjustments were made to the first submitted manuscript accordingly.

Figure R2 shows the motion recognition of the rigid-attached sensor and fabric-attached sensors (*i.e.*, F_2 - F_4) for prediction tasks of a variety of difficulties (*i.e.*, $\Delta\omega = \{0.3\pi, 0.47\pi, 0.57\pi, 0.72\pi, 0.82\pi \text{ rad s}^{-1}\}$) under both still air conditions and controlled airflow velocities (*i.e.*, 0.3 m/s and 0.47 m/s). The motion recognition performance when using fabric motion is influenced by airflow, particularly when the prediction task is challenging.

When the airflow velocity is low (*i.e.*, 0.3 m/s), fabric motion still shows enhanced motion recognition performance compared to rigid motion for challenging prediction tasks, although this advantage is less obvious compared to when external airflow is absent. Conversely, at a higher airflow velocity (*i.e.*, 0.47 m/s), the motion recognition performance of fabric motion is worse than that of rigid motion. This indicates that fabric motion is, unsurprisingly, susceptible to strong winds. For easy prediction tasks, there is no significant difference in the motion recognition performance between rigid and fabric motion. This suggests that external airflow speed has an impact on motion recognition

from fabric sensors, but the impact of this is small in less complex prediction tasks.

We also use the humidifier to generate different humidity levels and temperatures in the confined space and repeat the experiment to understand how they can affect the motion recognition performance of rigid and fabric motion.

Figure R3 shows the motion recognition of the rigid-attached sensor and fabric-attached sensors (*i.e.*, F_2 - F_4) for prediction tasks of a variety of difficulties (*i.e.*, $\Delta\omega = \{0.3\pi, 0.47\pi, 0.57\pi, 0.72\pi, 0.82\pi \text{rad s}^{-1}\}$) under varying humidity levels (*i.e.*, 45%, 55% and 65%). The result shows that varying humidity levels exhibit no significant impact on the performance of motion recognition using fabric motion.

Figure R4 shows the motion recognition of the rigid-attached sensor and fabric-attached sensors (*i.e.*, F_2 - F_4) for prediction tasks of a variety of difficulties (*i.e.*, $\Delta\omega = \{0.3\pi, 0.47\pi, 0.57\pi, 0.72\pi, 0.82\pi \text{rad s}^{-1}\}$) under varying room temperatures (*i.e.*, 15, 20 and 25 degree). The result also shows that varying room temperatures exhibit no obvious impact on the performance of motion recognition using fabric motion.

Changes:

We have added and discussed these results in Supplementary Information S4 (line 85-107).

C1.5. Comments to the authors

3. Compared to Aurora 6 degrees-of-freedom sensors, accelerators/inertial measurement units are more widely used. Please provide some experimental data to demonstrate that they still support the conclusions of this paper.

Our response

Thank you for this comment. While accelerometers and inertial measurement units (IMUs) are indeed widely used, their suitability depends on the specific application. In this study, we prioritise capturing fabric motion with *minimal influence on fabric dynamics*, since it is our view that, as technology develops, wearable sensors will become increasingly compact, lightweight, and integrated into natural clothing materials. Indeed, the latest electronic textiles already allow sensors to be embedded directly into clothing. For instance, Wicaksono et al. [3] successfully integrated accelerometer chips into fab-

Figure R1: Experimental set up for exploring the motion recognition performance of fabric motion under different environments.

Figure R2: Motion recognition accuracy from the initial time to various times up to 2.5 s comparison between rigid and fabrics under different wind speeds: (a) 0 m/s (b) 0.3 m/s (c) 0.47 m/s.

Figure R3: Motion recognition accuracy from the initial time to various times up to 2.5s comparison between rigid and fabrics under humidity levels: (a) 45% (b) 55% (c) 65%.

Figure R4: Motion recognition accuracy from the initial time to various times up to 2.5 s comparison between rigid and fabrics under room temperatures: (a) 15 (b) 20 (c) 25 degrees.

ric, making the sensor smaller and lighter, albeit this technology is not yet commercially available.

As introduced in [4], Jayasinghe et al attached conventional accelerometers (Actigraph tri-axial accelerometers (wGT3X-BT, weighing 19 g) to clothing. However, the weight of such sensors affects the movement of the fabric, influencing the experimental findings. The Aurora sensors used in our study are extremely lightweight and compact (0.92 mm × 9.4 mm), with negligible impact on fabric movement and therefore allowing us to eliminate potentially confounding effects (e.g., inertia, added fabric tension) that may be introduced with standard IMUs.

Moreover, as the Aurora sensors allow orientation measurements to be made, *they allow us to gather the same data as would be available using a standard IMU*. Several studies have demonstrated the interdependence between rotational angles and acceleration in motion analysis. Luinger et al. [5] showed that inclination angles derived from accelerometer data are functionally similar to direct rotation measurements, though the latter is generally more stable under dynamic conditions. Similarly, Zhou et al. [6] demonstrated that human motion tracking relies on both acceleration and orientation measurements, which can often be used interchangeably in various scenarios.

C1.6. Comments to the authors

4. Beyond the button press task, running scenarios can be added to further enhance the adaptability of the prediction model.

Our response

Thank you for your comment. The button press task requires participants to perform reaching motion. The choice of human reaching motions in this study was deliberately focused on fundamental, repetitive actions that form the basis of numerous daily tasks (e.g., object interaction, assistive robotics scenarios [7]). This approach allowed us to isolate and validate the core hypothesis regarding the effectiveness of fabric-mounted sensors under controlled, repeatable conditions—a critical step in establishing their feasibility compared to rigidly attached sensors. The simplicity of the reaching tasks was intentional to eliminate the confounding variables inherent in complex motions (e.g., ,

multi-joint coupling, environmental noise) and provide a standardised benchmark for comparing sensor performance in low-complexity scenarios, where traditional rigid sensors are often assumed to dominate. As demonstrated in the main manuscript, even in these simplified conditions, fabric-mounted sensors exhibit superior motion fidelity and comfort, challenging the conventional reliance on rigid attachments for basic movements.

C1.7. Comments to the authors

5. The motion of clothing contains additional information useful for analyzing human movement. By conducting experiments/simulations/theoretical modeling, the true relationship between the motion information of sensor-attached loose clothing (rather than a simple piece of fabric) and human movement can be quantified, further highlighting the importance of such additional information in motion prediction.

Our response

Thank you for your comment. We agree that conducting theoretical modelling is useful to understand the relationship between the motion information of sensor-attached loose clothing. Therefore, a probabilistic model explaining the observed phenomena is presented in Supplementary Information S2 (line 8-60), describing the relationship between the parameters of the sensorised fabric, the underlying (rigid body) movement, and the statistical distance as a measure of the accuracy that may be expected.

Changes:

We have added the theoretical modelling (*i.e.*, a probabilistic model) to Supplementary Information S2 (line 8-60).

Reviewer 2

C2.1. General Comment

The study presents interesting and innovative contributions, particularly regarding the use of loose garments with sensors to improve human movement recognition and prediction, challenging the traditional idea that sensors must be rigidly attached to the body. However, there are several methodological and validation issues that need significant improvements before the study can be considered for publication.

Our response

Thank you for your noting the novelty and general interest of our manuscript. Our point-by-point responses to the comments are given below.

C2.2. Comments to the authors

This study is based solely on empirical evidence, without a clear theoretical explanation or justification. Without a clear theoretical explanation, the results can seem incomplete or not robust because it's unclear why specific outcomes occur. This reduces the replicability or generalizability of the results. Generally, high-quality scientific journals do require a thorough discussion that not only presents results but also attempts to explain the phenomenon or offer hypotheses, at least preliminary, that can be tested in future studies. For example, a critical step would be to verify if the superior performance of fabric sensors is due to fabric noise being interpreted as useful signals.

Our response

Thank you for your comment. We agree that providing a theoretical framework alongside the empirical results is important for the robustness and generalisability of our findings.

To address the concern, we have introduced a probabilistic model to explain the phenomenon observed. Specifically, we demonstrate that statistical distance measures, such as Kullback-Leibler divergence, imply that stochastic fabric movements lead to greater

discriminative ability. This model provides a theoretical foundation for the enhanced performance of fabric sensors, particularly when sensors are loosely attached to the fabric as opposed to being rigidly attached. We believe these additions strengthen the theoretical foundation of our study and help to explain the observed outcomes more clearly.

Changes:

We have added the probabilistic model of fabric motion to Supplementary Information (S2 line 8-60).

C2.3. Comments to the authors

To do this, the authors could (a) conduct experiments where the fabric is fixed to limit oscillations while keeping body movement constant. If accuracy decreases significantly, it could indicate that fabric oscillations were providing additional (possibly noisy) information.

Our response

We thank you for your suggestion. In response, we conducted experiments in which the fabric was fixed at the distal end to a stationary rigid object (*i.e.*, that does not move with the yoke) to limit oscillations at the tip of the fabric. The sensor placements and experimental setup remain otherwise unchanged. Figure R5(a) (below) shows the experimental setup.

Figure R5(b) and (c) show the motion recognition accuracy for the four sensors when the tip of the fabric has unlimited and limited oscillations, respectively. As can be seen, the accuracy drops for all sensors under these conditions, especially F_4 . This suggests that the fabric oscillations play a role in providing useful additional information to help motion recognition and prediction, as hypothesised in the comment, supporting the findings reported in the manuscript.

C2.4. Comments to the authors

Figure R5: (a) The experimental setup to limit fabric oscillations, where the fabric is fixed to a rigid object. (b) Motion recognition accuracy from the initial time to various times up to 2.5s without and (c) with limiting oscillations.

(b) introduce artificial noise to the data to see if the model's performance decreases. If the model continues to perform well despite added noise, it suggests the system is robust and not dependent on random artifacts.

Our response

Thank you for your comment. We agree that adding artificial noise can help test the robustness of the findings. Therefore, we use the experiment involving a scotch yoke with a piece of fabric attached as an example.

For this, we set up a numerical simulation of the scotch yoke moving at angular velocities $\omega = \{0.63\pi, 0.75\pi, 1.2\pi, 1.45\pi \text{ rad s}^{-1}\}$. Motion recognition is made between the case when the scotch yoke moves at the lowest frequency (*i.e.*, $\omega_1 = 0.63\pi \text{ rad s}^{-1}$) and each higher frequency (*i.e.*, $\omega_2 = \{0.75\pi, 1.2\pi, 1.45\pi, \text{ rad s}^{-1}\}$), using data samples of the rigid motion R and artificial fabric motion F . The artificial fabric motion is simulated as follows.

According to the proposed probabilistic model in supplementary information S1, we assume that the random offset caused by fabric motion follows a Gaussian distribution, with the variance of this distribution increasing as the scotch yoke moves faster. The orientation of the fabric movement, y^f (*i.e.* $\mathcal{M} = 1$), of the point p_y^f on the fabric at any given time is modelled as a stochastic process consisting of the corresponding orientation, y^r (*i.e.* $\mathcal{M} = 1$), of a point p_y^r on the rigid body, plus a random offset, δ , introduced by the fabric motion:

$$X^f = X^r + \Delta, \quad (1)$$

where $y^f(t) \sim X^f$, $y^r(t) \sim X^r$, and $\delta(t) \sim \Delta$, and $\Delta \sim \mathcal{N}(\mu, \sigma^2)$. Based on the results of maximum likelihood estimation from Supplementary Information 1, the standard deviation (σ^2) in the Gaussian distribution for the random offset is 0.1678, 0.2238, 0.4231, 0.9622 when the scotch yokes move at the aforementioned frequencies.

Figure R6(a,b) presents comparative analyses of motion recognition accuracy from the initial time to various times up to 2.5 s for yoke frequency differences $\Delta\omega = \{0.12\pi, 0.53\pi, 0.82\pi\} \text{ rad s}^{-1}$, utilising experimentally acquired data and synthetic

Figure R6: The motion recognition accuracy from the initial time to various times up to 2.5 s for a yoke frequency difference of $\Delta\omega = \{0.12\pi, 0.53\pi, 0.82\pi\} \text{ rad s}^{-1}$ using real and artificial data generated by the proposed probabilistic model.

datasets generated through our probabilistic modelling framework (see Supplementary Information S2). Crucially, the system maintains superior classification accuracy (*i.e.*, R_1) even under artificially introduced noise conditions. These findings ensure the robustness of this system.

C2.5. Comments to the authors

the authors could analyze the noise present in the data collected from fabric-mounted sensors. Tools like Fourier transforms or noise filtering techniques (e.g., Kalman filters) could help distinguish fabric oscillations caused by body movement from unrelated oscillations.

Our response

Our response

Thank you for your valuable comment. We appreciate your suggestion regarding the noise analysis. In response, we have indeed utilised Fourier transforms to examine the noise in the data from fabric-mounted sensors, as demonstrated in Figure R7(a). The noise is computed using equation 1. The Fourier transform results clearly show distinct frequency components (such as $\omega = \{0.63\pi, 1.2\pi, 1.45\pi \text{rad s}^{-1}\}$). As depicted in the figure, there are significant differences in the amplitudes of various frequency components. The amplitude corresponding to $1.45\pi \text{rad s}^{-1}$ is distinctly higher, indicating a stronger energy at this frequency component within the data, which likely represents the dominant oscillation frequency caused by body movement. In contrast, frequencies like $0.63\pi \text{rad s}^{-1}$ have relatively lower amplitudes, implying weaker energy. By analysing these frequency components, we can distinguish fabric oscillations caused by body movement from unrelated oscillations. The prominent peaks in the amplitude frequency plot (e.g., the high amplitude peak for $1.45\pi \text{rad s}^{-1}$ help identify the dominant frequencies associated with specific types of oscillations. Unrelated oscillations generally do not display such marked high amplitudes at these specific frequencies, allowing us to isolate meaningful body - movement - related oscillations from complex data. This analysis using Fourier transforms has effectively aided in differentiating the relevant and irrelevant oscillations in the data, reinforcing the solid foundation our current Fourier transform based analysis provides for distinguishing the oscillations as shown in the figure.

We also apply a low-pass filter to the noise to understand fabric oscillations. A fourth-order Butterworth low-pass filter was used to process the noise, with a cutoff frequency set at 5 Hz and a sampling frequency of $f_s = 40$ Hz. This configuration was designed to attenuate frequency components above 5 Hz, thereby retaining the lower-frequency signals while reducing high-frequency noise. It is clear that the range of artefacts is bigger when the Scotch Yoke runs faster.

C2.6. Comments to the authors

In order to avoid an intrinsic bias of the LR-HMM model, it could be compared with other advanced models, such as Recurrent Neural Networks (RNNs) or Long Short-Term Memory (LSTM) networks to confirm that the improvement is not due to bias in the

Figure R7: Analysis of noise and signal filtering in fabric-mounted sensor data using (a) Fourier transforms and (b) low pass filter.

HMM model.

Our response

We thank you for your comment and recognise the importance of addressing potential model bias. As such, we experimented with other advanced models, specifically Long Short-Term Memory (LSTM) networks. We adopt a systematic tuning strategy: we first compare the effects of combining different hidden layer dimensions (32/64/128), learning rates (0.001–0.01 gradient), and training periods (50–200 epochs) on the validation set via Grid Search. The LSTM model was finally configured with an input dimension of 3 (to handle the three axes of orientation movement), a hidden layer size of 64, and trained for 100 epochs using the Adam optimiser at a learning rate of 0.005.

Figure R8 presents the motion recognition accuracy for classifying the difference in yoke movement, $\Delta\omega = 0.12\pi\text{rads}^{-1}$, using two different algorithms. Figure R8(b) shows the motion recognition accuracy using LSTM. As can be seen, the LSTM did not outperform the LR-HMM (see Figure R8(a)) in terms of motion recognition accuracy. On the contrary, the LR-HMM provided higher motion recognition accuracy overall.

Despite the fact that LSTMs are well-suited for sequential data and have shown impressive results in various tasks, we hypothesise that the superior performance of the LR-HMM in this context may stem from two inherent advantages of Hidden Markov

Figure R8: The motion recognition accuracy from the initial time to various times up to 2.5 s for a yoke frequency difference of $\Delta\omega = 0.12\pi\text{rad s}^{-1}$ using two machine learning algorithms (a) Left-to-Right Hidden Markov Model and (b) Long Short-Term Memory.

model (HMM) architectures: (a) their demonstrated data efficiency through optimal exploitation of temporal dependencies in sparse training sets, and (b) significantly reduced parametric complexity compared to deep learning alternatives, enabling robust convergence with limited samples. The LR-HMM is particularly advantageous when limited training data is available, as it requires fewer parameters and can efficiently capture temporal dependencies in motion data.

Nonetheless, the results obtained from the LSTM model still support our findings that fabric-attached sensors lead to higher motion recognition accuracy and can achieve satisfactory results more quickly than other sensor types. This suggests that, regardless of the statistical learning approach, the use of loosely attached sensors is beneficial to motion analysis.

Changes:

We have added the results of using the LSTM model and discussed these results in the Supplementary Information (S3 lines 61–84).

C2.7. Comments to the authors

The authors could conduct a correlation analysis to see if fabric oscillations are tightly correlated with body movements. Measure how synchronized fabric movement (recorded by the fabric-mounted sensors) is with actual body movement. If there's a strong correlation, it would indicate that the fabric is amplifying body movement in a useful way. If correlation is low, it's possible that the improvement is due to noise artefacts.

Our response

We appreciate your suggestions. To investigate this question, Pearson correlation coefficients were calculated between rigid body motion and the movement of each fabric-mounted sensor while the Scotch yoke mechanism was operated at different frequencies. As summarised in Table R2, the resulting correlation coefficients were consistently low (approaching 0) across all frequencies.

However, we respectfully contest the assertion that low correlation between fabric oscillations and rigid body kinematics renders our findings “artefacts of noise”. This interpretation fundamentally misrepresents both the intricate dynamics of textile behaviour and the operational characteristics of sensorised garments. Fabric motion cannot be simplistically conceptualised as a linear proxy for bodily movement, but rather emerges from complex non-linear interactions between the biomechanics of the body forces and intrinsic textile properties. It is these sophisticated material responses that enhance the motion recognition capabilities of fabric-based systems.

C2.8. Comments to the authors

Using different fabrics with varying physical properties (e.g., rigid vs. flexible fabrics) could help better understand the role fabric plays in improving performance. If sensors on more rigid fabrics do not show improvement compared to flexible ones, it could confirm that flexible fabric movement is useful for motion recognition.

Our response

We thank you for your suggestion. It is useful to understand whether using different fabrics with varying physical properties can improve motion recognition performance. To address this question, we conducted controlled repetitions of the Scotch yoke ex-

Frequency of Scotch Yoke ($\pi\text{rad s}^{-1}$)	F_2	F_3	F_4
0.63	0.23	0.2	-0.28
0.75	0.17	0.19	-0.21
0.92	0.13	0.14	-0.15
1.1	0.03	0.17	-0.15
1.2	-0.02	-0.05	-0.11
1.35	0.02	0.02	-0.01
1.45	0.0110	-0.004	-0.04

Table R2: Correlation values between rigid motion and the movement of each fabric-attached sensor when the scotch yoke runs at different frequencies.

periment with two different types of stretchable fabrics attached (*i.e.*, denim and woven cotton). Motion recognition between two different frequencies of the Scotch yoke $\Delta\omega = 0.3\pi\text{rad s}^{-1}$ is made. Denim is far more stretchable than woven cotton. Figure R9 (a) and (b) show the motion recognition accuracy of rigid and fabric motion under this experimental setup, with denim and woven cotton attached, respectively. This result shows that using more stretchable fabric can improve motion recognition performance.

C2.9. Comments to the authors

the experiments were conducted in extremely controlled environments (e.g., with robotic devices or a scotch yoke), which may not reflect the complexities of real-world situations. The authors themselves acknowledge that external environmental factors (such as wind, temperature, and humidity) were not considered. It would be helpful to include experiments or simulations that consider more realistic contexts, perhaps in outdoor environments or with varying environmental factors. This could help demonstrating that the system is applicable beyond the lab.

Our response

Thank you for your comment. We agree that investigating the impact of environmental conditions on the findings is important. Therefore, we have added more experiments to investigate this question. Please refer to C1.4 for more details.

Figure R9: The motion recognition accuracy from the initial time to various times up to 2.5 s for a yoke frequency difference of $\Delta\omega = 0.12\pi\text{rad s}^{-1}$ when (a) denim and (b) woven cotton are attached.

C2.10. Comments to the authors

There's a human-reaching experiment in which the two types of sensors, mounted on the body and on fabric, were compared. Also in this case, although the experiment outcomes are very promising results, the tested reaching task is relatively simple and predictable. More complex movements, such as 3D or irregular movements, were not considered. It would be necessary to test the system on more complex and dynamic movements to verify whether fabric-mounted sensors maintain their superiority.

Our response

We appreciate your constructive feedback and acknowledge the positive remarks. The choice of human reaching motions in this study was deliberately focused on fundamental, repetitive actions that form the basis of numerous daily tasks (*e.g.*, object interaction, assistive robotics scenarios [7]). This approach allows us to isolate and validate the core hypothesis regarding the effectiveness of fabric-mounted sensors under controlled, repeatable conditions—a critical step in establishing their feasibility compared to rigidly attached sensors. The simplicity of this human reaching tasks was intentional to eliminate the confounding variables inherent in complex motions (*e.g.*, , multi-joint coupling) and provide a benchmark for comparing sensor performance in low-complexity scenar-

ios, where traditional rigid sensors are often assumed to dominate. As demonstrated in the main manuscript, even in these simplified conditions, fabric-mounted sensors exhibited superior motion fidelity and comfort, challenging the conventional reliance on rigid attachments for basic movements.

However, we respectfully contest the assertion that human reaching motion in the manuscript is 3D movement. While irregular movements were not explicitly tested in this work, the study's primary contribution lies in pioneering the use of fabric-integrated sensors for motion capture, which addresses long-standing issues with rigid sensors (*e.g.*, discomfort, limited flexibility), and establishing a baseline performance metric for fabric-mounted sensors in fundamental motions. This baseline serves as a reference for future comparative studies in more complex scenarios. The manuscript notes that the observed advantages in simple motions suggest strong potential for fabric-mounted sensors in applications such as augmented reality (AR) gesture tracking or rehabilitation robotics, where both comfort and motion accuracy are critical.

Exploring sensor performance under dynamic environments would be interesting. We repeated the experiments under varying conditions using a Scotch yoke experimental setup. Wind speed influences the motion recognition performance of clothing motion: higher wind speeds correlate with worse motion recognition performance. By contrast, room temperature and humidity levels had no discernible effect. For more detailed experimental results, please refer to C1.4. These findings support the premise that fabric-mounted sensors can be a valuable tool in motion analysis, especially in low-wind settings (such as indoors).

C2.11. Comments to the authors

[continued] Again, the experiment was conducted in a highly controlled environment, with known position targets and a fixed setup. In real-world contexts, where external interferences (*e.g.*, wind moving the fabric) could occur, the results might change. Conducting experiments in less controlled environments would help to assess how the system behaves with additional environmental variables.

Our response

We thank you for your comment. We have investigated the motion recognition performance of fabric motion under different environmental conditions (*e.g.*, wind speeds, room temperature, and humidity levels). We repeated the experiments under varying conditions using a Scotch yoke experimental setup. Wind speed significantly influences the motion recognition performance of clothing motion: higher wind speeds correlate with worse motion recognition performance. By contrast, room temperature and humidity levels had no discernible effect. For more detailed experimental results, please refer to C1.4.

humidity was deliberate to avoid conflating their potential effects on human posture (as hypothesised) with the primary focus of fabric motion analysis.

C2.12. Comments to the authors

[continued] Although the study includes 22 participants, it would be advisable to test the system on a larger and more diverse sample, including different types of movement, execution styles, and physical characteristics of the participants. More natural and less standardized movements could present challenges for the system, especially if the fabric behaves unpredictably.

Our response

We appreciate your thoughtful suggestion. The current study focused on a controlled sample of 22 participants, a size justified by a priori power analysis for repeated measures ANOVA. For a medium effect size (0.3) and statistical power of 0.9, this sample size meets the requirements outlined in [8], ensuring robust detection of differences between sensor types in standardised movements.

The invited participants exhibited remarkable diversity, encompassing a wide spectrum of anthropometric and laterality characteristics. The cohort comprised fourteen males and eight females, with significant variability in physical attributes: mean \pm SD values included age 25.2 ± 3.5 years, height 174 ± 7.4 cm, body mass 74 ± 17 kg, arm length 51 ± 3.5 cm, and wrist circumference 16.5 ± 1.7 cm, reflecting substantial individual differences in stature and body composition. In terms of laterality, assessment via

the Edinburgh Handedness Questionnaire [9] further highlighted the cohort’s diversity. This combination of heterogeneous anthropometric traits and nuanced laterality profiles underscores the robust diversity within the invited sample, ensuring a comprehensive representation of individual variation for research purposes.

C2.13. Comments to the authors

The use of LR-HMMs, combined with the forward-backward algorithm and Viterbi algorithm, can be computationally expensive, especially when increasing the number of states. It is unclear whether the proposed method is scalable to large datasets or applicable in real-time, especially in applications like virtual reality or human-robot collaboration, as suggested. The authors should provide an analysis of computational complexity and discuss how the system would perform on larger datasets or in real-time applications.

Our response

We understand the concern regarding the computational complexity of LR-HMMs. In our approach, during the deployment stage, the computational complexity depends on the number of hidden states \mathcal{N} , the dimensionality of the data \mathcal{M} , and the number of time steps v , as follows.

The first step is motion recognition, which is performed using the forward algorithm. The complexity of this algorithm is [10]

$$O(\mathcal{N}^2 v \mathcal{M}). \quad (2)$$

In the experiments reported in the manuscript, the hardware configuration consists of a Intel Core i7-8700 @3.20GHz \times 12 CPU with 16GB memory, running MATLAB 2022b, for which the processing time is approximately 0.0035 s.

The next step in the process is motion prediction, which relies on the Viterbi algorithm. The Viterbi algorithm is used to find the most likely sequence of hidden states given the observed trajectory. The computational complexity of the Viterbi algorithm is similar to that of the forward algorithm, as it also involves computing probabilities over all hidden states at each time step. However, in this case, we need to perform an additional

operation to keep track of the most likely path. The complexity of this algorithm is [10]

$$O(\mathcal{N}^2\mathcal{V}\mathcal{M}), \quad (3)$$

where \mathcal{V} is the total number of time steps. The processing time for this step is approximately 0.012 s.

Our LR-HMM implementation therefore has 15.5 ms total processing time (3.5 ms for recognition + 12 ms for prediction). This meets the stringent latency requirements of virtual reality systems, where smaller than 20 ms response times are critical for maintaining immersion and preventing motion sickness [11]. For robotic systems that require high-precision collaboration, the delay time should typically be less than 50 ms to ensure precise synchronisation of movements and reduce system reaction time [12].

We acknowledge that extreme-scale deployments (*e.g.*, > 500 states) would require further architectural innovations, which we are pursuing through collaborative edge-cloud computing frameworks in ongoing work.

Changes:

We have added this computational complexity computation to Supplementary Information S5 (line 108-135).

References

- [1] Ingmar Visser. Seven things to remember about hidden Markov models: A tutorial on Markovian models for time series. *Journal of Mathematical Psychology*, 55(6):403–415, 2011.
- [2] Xanthi S Papageorgiou, Georgia Chalvatzaki, Costas S Tzafestas, and Petros Maragos. Hidden markov modeling of human normal gait using laser range finder for a mobility assistance robot. In *2014 IEEE International Conference on Robotics and Automation (ICRA)*, pages 482–487. IEEE, 2014.
- [3] Irmandy Wicaksono, Carson I Tucker, Tao Sun, Cesar A Guerrero, Clare Liu, Wesley M Woo, Eric J Pence, and Canan Dagdeviren. A tailored, electronic textile conformable

- suit for large-scale spatiotemporal physiological sensing in vivo. *npj Flexible Electronics*, 4(1):1–13, 2020.
- [4] Udeni Jayasinghe, Faustina Hwang, and William S Harwin. Comparing loose clothing-mounted sensors with body-mounted sensors in the analysis of walking. *Sensors*, 22(17):6605, 2022.
- [5] Henk J Luinge and Peter H Veltink. Inclination measurement of human movement using a 3-d accelerometer with autocalibration. *IEEE Transactions on neural systems and rehabilitation engineering*, 12(1):112–121, 2004.
- [6] Huiyu Zhou and Huosheng Hu. Human motion tracking for rehabilitation—a survey. *Biomedical signal processing and control*, 3(1):1–18, 2008.
- [7] Amir Haddadi and Keyvan Hashtrudi-Zaad. Robust stability of teleoperation systems with time delay: A new approach. *IEEE Transactions on Haptics*, 6(2):229–241, 2013.
- [8] Franz Faul, Edgar Erdfelder, Axel Buchner, and Albert-Georg Lang. Statistical power analyses using G* Power 3.1: Tests for correlation and regression analyses. *Behavior research methods*, 41(4):1149–1160, 2009.
- [9] Richard C Oldfield. The assessment and analysis of handedness: the Edinburgh inventory. *Neuropsychologia*, 9(1):97–113, 1971.
- [10] Lawrence R Rabiner. A tutorial on hidden Markov models and selected applications in speech recognition. *Proceedings of the IEEE*, 77(2):257–286, 1989.
- [11] Kjetil Raaen and Ivar Kjellmo. Measuring latency in virtual reality systems. In *Entertainment Computing-ICEC 2015: 14th International Conference, ICEC 2015, Trondheim, Norway, September 29-October 2, 2015, Proceedings 14*, pages 457–462. Springer, 2015.
- [12] Christiane Attig, Nadine Rauh, Thomas Franke, and Josef F Krems. System latency guidelines then and now—is zero latency really considered necessary? In *Engineering Psychology and Cognitive Ergonomics: Cognition and Design: 14th International Conference, EPCE 2017, Held as Part of HCI International 2017, Vancouver, BC, Canada, July 9-14, 2017, Proceedings, Part II 14*, pages 3–14. Springer, 2017.

Reply to the review report on

Can your Shirt Predict your Movement? Improving Motion Analysis using Loose Clothing

by Tianchen Shen, Sacha Morris, Irene Di Giulio and Matthew Howard

August 30, 2025

Thank you for giving us the opportunity to submit a revised draft of the manuscript “Can your Shirt Predict your Movement? Improving Motion Analysis using Loose Clothing” for publication in *Nature Communications*.

We would like to thank the reviewers for reviewing our paper again. We have addressed all the comments and updated the manuscript. In this response letter, the reviewers’ original comments are highlighted with a red background, our replies appear in blue background, and the amendments made to the manuscript are enclosed in quotation marks and distinguished by a green background. The changes made in the manuscript are marked as red. For ease of reference, we have assigned a unique number to each comment in the form CX.Y, where X refers to the reviewer number and Y to the comment of that reviewer. Below, we provide detailed responses to the reviewers’ comments.

Reviewer 1

C1.1. Comments to the authors

The authors' responses present a number of clear strengths. First, they provide a well-articulated theoretical explanation for the observed benefits of fabric-based sensing. The probabilistic model introduced in the supplementary material, grounded in the concept of Kullback-Leibler divergence, offers a solid and intuitive rationale for why motion recognition improves when using loosely attached sensors. This modeling work is both insightful and well-aligned with the experimental results.

In addition, the authors undertook significant new experimental work to test their approach under variable environmental conditions. The simulations of airflow, humidity, and temperature are particularly valuable in demonstrating the robustness of the method in more realistic settings. These tests confirm that, under typical conditions, the fabric sensors perform reliably, and in some cases, better than rigid sensors. Only at higher wind speeds does their performance start to decline. Furthermore, the authors show that their findings are not tied to a single statistical approach by implementing and tuning an LSTM model, which, although less effective than the LR-HMM, still supports the main claims. This strengthens the argument that the improvement is due to the nature of the sensing rather than to the specifics of the model used.

Our response

We would like to thank you for your feedback, which has improved the quality of our manuscript.

C1.2. Comments to the authors

Nonetheless, there are still some important issues that remain partially addressed. The comparison of learning models, while welcome, is limited to just one alternative (LSTM) and is confined to a single experimental scenario. A broader analysis involving additional models, such as GRUs or temporal convolutional networks, and across different types of movement would provide stronger evidence of generalizability.

Our response

Thank you for your comment. We appreciate the suggestion to expand the comparison of learning models, which has prompted us to conduct additional experiments with transformer neural networks (TNNs) and temporal convolutional network (TCN) to validate the generalisability of our findings further. We report these additional experiments in Supplementary Information S3, where it can be observed that the effects of using data from the loose fabric are broadly similar to those seen when using the left-to-right hidden Markov model (LR-HMM).

Changes:

We have added and discussed these statistical results in Supplementary Information S3.

C1.3. Comments to the authors

[continued] More importantly, the authors did not provide any validation using standard IMUs. While they justify their focus on textile-embedded sensors as a future direction, IMUs remain the current standard in many fields, and even a minimal empirical comparison would have strengthened the practical credibility of their claims.

Our response

We thank you for your comment. To examine whether inertial measurement units (IMUs) can support the conclusions presented in this paper, we repeated the scotch yoke experiment, replacing the Aurora 6 degrees-of-freedom sensors with IMUs. One ADXL335 sensor (Analog Devices, Inc., USA) was attached to the yoke to capture rigid body motion, and a second sensor was affixed to the fabric. The presence of the IMU and jumper wires may influence fabric motion. To minimise this effect, only one sensor was attached to the fabric F , as opposed to the three Aurora sensors used in the main manuscript. The motion recognition performance for both rigid and fabric motion was evaluated using the three-axis acceleration data from the sensors.

We report the motion recognition accuracy of the rigid-attached sensor and fabric-attached sensors for the prediction task (*i.e.*, $\Delta\omega = 0.3\pi\text{rad s}^{-1}$) in Supplementary Information S6. It can be found that the motion recognition accuracy of fabric motion is still significantly higher than using rigid motion. This proves that using IMUs can also

support our findings.

Changes:

We have added and discussed the result in Supplementary Information S6.

C1.4. Comments to the authors

Finally, although the environmental experiments are noteworthy, the lack of formal statistical tests (e.g., ANOVA or similar) somewhat limits the strength of the conclusions regarding robustness. Including such analysis would help quantify how much environmental variation truly impacts recognition performance.

Our response

We thank you for your comment. Regarding your suggestion to supplement statistical tests to enhance the robustness of our conclusions, we have conducted further analyses as follows.

We assessed the distribution of motion recognition accuracy of fabric motion across different environments from the initial time to various times up to 2.5 s. Normality tests (specifically the Kolmogorov-Smirnov test) indicated that the data deviate from a normal distribution ($P < 0.05$). As parametric tests such as ANOVA rely strongly on the assumption of normality [1], we were mindful of their limitations in this context.

While nonparametric alternatives like the Wilcoxon signed-rank test are well-suited to non-normal data (and applicable to large samples), we noted that this test is highly sensitive to trivial differences under large sample sizes (100 per group). Even a negligible difference between two groups may achieve statistical significance without practical relevance, potentially misleading interpretation.

To illustrate this, We take the motion accuracy for each when the room temperature is 15, 20 degree as an example, with the results shown in Figure R1. Figure R1 shows these results. As observed in the figure, the motion accuracy of the fabric sensor varies under different room temperatures. The motion accuracy of the fabric sensor behaves differently under different room temperatures. For instance, the motion accuracy of sensor F_4 at 0.25 s is 100% at 15 and 95% at 20 degree. Although these values differ noticeably, the improvement in accuracy over sensor R_1 remains comparable. We consider

Figure R1: Trivial differences in fabric motion recognition performance at different temperatures (*i.e.*, 15 and 20 degrees).

this difference trivial, indicating no practical significance, though it could be statistically significant. From the perspective of the machine learning methods employed, such minor differences could stem from either the selection of distinct trajectories for the training/testing sets or the use of varying initial parameters in the implementation of LR-HMM. Importantly, these factors do not affect the reproducibility of our findings.

Therefore to understand *how much* environmental variation truly impacts recognition performance (rather than merely establishing whether a difference exists), we therefore opted to conduct an effect size analysis. This method quantifies the gap between groups relative to their variability in a standardised way. For this, we use Cohen's d , as a robust metric for evaluating effect sizes even in non-normally distributed data [2].

This analysis is provided in Supplementary Information S4, where it can be seen that the Cohen's d values are consistent with our previous findings.

Changes:

We have added and discussed these statistical results in Supplementary Information S4.

Reviewer 2

C2.1. Comments to the authors

The authors have thoroughly revised the manuscript to address the concerns raised. The quality of the manuscript has significantly improved and is now suitable for publication consideration.

Our response

We would like to thank you for your positive feedback and for considering our manuscript for publication.

References

- [1] Andy Field. *Discovering statistics using IBM SPSS statistics*. Sage publications limited, 2024.
- [2] Geoff Cumming. *Understanding the new statistics: Effect sizes, confidence intervals, and meta-analysis*. Routledge, 2013.

Reply to the review report on

Human motion recognition and prediction using loose cloth

by Tianchen Shen, Sacha Morris, Irene Di Giulio and Matthew Howard

November 11, 2025

Thank you for accepting our manuscript, “Human motion recognition and prediction using loose cloth”, for publication in *Nature Communications*, in principle.

We would like to thank the reviewers for reviewing our paper again. We have addressed all the comments and updated the manuscript. In this response letter, the reviewers’ original comments are highlighted with a red background, our replies appear in blue background, and the amendments made to the manuscript are enclosed in quotation marks and distinguished by a green background. The changes made in the manuscript are marked as red. For ease of reference, we have assigned a unique number to each comment in the form CX.Y, where X refers to the reviewer number and Y to the comment of that reviewer. Below, we provide detailed responses to the reviewers’ comments.

Reviewer 1

C1.1. Comments to the authors

The revised manuscript has been significantly improved. The addition of TNN and TCN, the IMU proof-of-concept, and the expanded statistical analyses and experiments clearly strengthen the paper. The manuscript is now clear and well-structured.

Before acceptance, I'd only ask for two minor clarifications. Please make it explicit that the IMU experiment is a proof-of-concept and that a more systematic validation is left for future work. It would help to include a short note in the Discussion acknowledging the limits of the current statistical analysis and why more complex methods weren't applied here.

Our response

We sincerely thank the reviewer for this constructive suggestion.

We have added a new paragraph to the Discussion section that explicitly addresses both points. The new text clearly frames the IMU experiment as a preliminary proof-of-concept and acknowledges that the current statistical analysis supporting the IMU applicability is constrained by the simplicity of this initial validation. We also state that more extensive experiments and stronger statistical evaluations are required in future work to substantiate the findings for IMU-based systems.

We believe this addition provides the requested clarifications and accurately sets the context for the current findings and future research directions.

Changes:

We have added a new paragraph in the Discussion section (Line 268-274).